# SAMPLE-EFFICIENT LEARNING OF POMDPS WITH MULTIPLE OBSERVATIONS IN HINDSIGHT

**Jiacheng Guo**[*]  **Minshuo Chen**[*]  **Huan Wang**[†]  **Caiming Xiong**[†]  **Mengdi Wang**[*]  **Yu Bai**[†]
[*]Princeton University    [†]Salesforce Research

## ABSTRACT

This paper studies the sample-efficiency of learning in Partially Observable Markov Decision Processes (POMDPs), a challenging problem in reinforcement learning that is known to be exponentially hard in the worst-case. Motivated by real-world settings such as loading in game playing, we propose an enhanced feedback model called "multiple observations in hindsight", where after each episode of interaction with the POMDP, the learner may collect multiple additional observations emitted from the encountered latent states, but may not observe the latent states themselves. We show that sample-efficient learning under this feedback model is possible for two new subclasses of POMDPs: *multi-observation revealing POMDPs* and *tabular distinguishable POMDPs*. Both subclasses generalize and substantially relax *revealing POMDPs*—a widely studied subclass for which sample-efficient learning is possible under standard trajectory feedback. Notably, distinguishable POMDPs only require the emission distributions from different latent states to be *different* instead of *linearly independent* as required in revealing POMDPs.

## 1 INTRODUCTION

Partially observable reinforcement learning problems, where the agent must make decisions based on incomplete information about the environment, are prevalent in practice, such as robotics (OpenAI et al., 2019), economics (Zheng et al., 2020) and decision-making in education or clinical settings (Ayer et al., 2012). However, from a theoretical standpoint, it is well established that learning a near-optimal policy in Partially Observable Markov Decision Processes (POMDPs) requires exponentially many samples in the worst case (Mossel & Roch, 2005; Krishnamurthy et al., 2016). Such a worst-case exponential hardness stems from the fact that the observations need not provide useful information about the true underlying (latent) states, prohibiting efficient exploration. This is in stark contrast to fully observable RL in MDPs in which a near-optimal policy can be learned in polynomially many samples, without any further assumption on the MDP (Kearns & Singh, 2002; Auer et al., 2008; Azar et al., 2017).

Towards circumventing this hardness result, one line of recent work seeks additional structural conditions under which a polynomial sample complexity is possible (Katt et al., 2018; Liu et al., 2022a; Efroni et al., 2022). A prevalent example there is revealing POMDPs (Jin et al., 2020a; Liu et al., 2022a), which requires the observables to reveal *some* information about the true latent state so that the latent state is (probabilistically) distinguishable from the observables. Another approach, which we explore in this paper, entails using enhanced feedback models that deliver additional information beyond what is provided by standard trajectory-based feedback. This is initiated by the work of Lee et al. (2023), who proposed the framework of *Hindsight Observable POMDPs* (HOMDPs). In this setting, latent states are revealed in hindsight *after each episode has finished*. This hindsight revealing of latent states provides crucial information to the learner, and enables the adaptation of techniques for learning fully observable MDPs (Azar et al., 2017). As a result, it allows a polynomial sample complexity for learning any POMDP (tabular or with a low-rank transition) under this feedback model, negating the need for further structural assumptions (Lee et al., 2023).

In this paper, we investigate a new feedback model that reveals *multiple additional observations*—emitted from the same latent states as encountered during each episode—in hindsight to the learner. As opposed to the hindsight observable setting, here the learner does not directly observe the latent states, yet still gains useful information about the latent states via the additional observations. This model resembles practical scenarios such as the save/load mechanism in game playing, in which

the player can replay the game from a previously saved state. Similar feedback models such as RL with replays (Amortila et al., 2022; Li et al., 2021; Lee et al., 2023) have also been considered in the literature in fully observable settings. This feedback model is also theoretically motivated, as the additional observations in hindsight provide more information to the learner, which in principle may allow us to learn a broader class of POMDPs than under standard feedback as studied in existing work (Jin et al., 2020a; Liu et al., 2022a; Zhan et al., 2022; Chen et al., 2022a; Liu et al., 2022b).

Our contributions can be summarized as follows.

- We define a novel feedback model—POMDPs with $k$ multiple observations ($k$-MOMDP)—for enhancing learning in POMDPs over the standard trajectory feedback (Section 3). Under $k$-MOMDP feedback, after each episode is finished, the learner gains additional observations emitted from the same latent states as those visited during the episode.

- We propose *$k$-MO-revealing POMDPs*, a natural relaxation of revealing POMDPs to the multiple observation setting, and give an algorithm ($k$-OMLE) that can learn $k$-MO-revealing POMDPs sample-efficiently under $k$-MOMDP feedback (Section 4). Concretely, we provide learning results for both the tabular and the low-rank transition settings.

- We propose *tabular distinguishable POMDPs* as an attempt towards understanding the minimal structural assumption for sample-efficient learning under $k$-MOMDP feedback (Section 5.1). While being a natural superset of $k$-MO-revealing POMDPs for all $k$, we also show a reverse containment that any distinguishable POMDP is also a $k$-MO-revealing POMDP with a sufficiently large $k$. Consequently, any distinguishable POMDP can be learned sample-efficiently by reducing to $k$-MO-revealing POMDPs and using the $k$-OMLE algorithm (Section 5.2).

- For distinguishable POMDPs, we present another algorithm OST (Section 5.3) that achieves a sharper sample complexity than the above reduction approach and is computationally efficient given a POMDP planning oracle. The algorithm builds on a closeness testing subroutine using the multiple observations to infer the latent state up to a permutation. Technically, compared with the reduction approach whose proof relies *implicitly* on distribution testing results, the OST algorithm utilizes distribution testing techniques *explicitly* in its algorithm design.

## 1.1 RELATED WORK

**Sample-efficient learning of POMDPs**   Due to the non-Markovian characteristics of observations, policies in POMDPs generally rely on the complete history of observations, making them more challenging to learn compared to those in fully observable MDPs. Learning a near-optimal policy in POMDPs is statistically hard in the worst case due to an exponential sample complexity lower bound in the horizon (Mossel & Roch, 2005; Krishnamurthy et al., 2016; Papadimitriou & Tsitsiklis, 1987), and also computationally hard (Papadimitriou & Tsitsiklis, 1987; Vlassis et al., 2012).

To circumvent this hardness, a body of work has been dedicated to studying various sub-classes of POMDPs, such as revealing POMDPs (Hsu et al., 2012; Guo et al., 2016; Jin et al., 2020c; Liu et al., 2022a; Chen et al., 2023), and decodable POMDPs (Efroni et al., 2022) (with block MDPs (Krishnamurthy et al., 2016; Du et al., 2019; Misra et al., 2020) as a special case). Other examples include reactiveness (Jiang et al., 2017), revealing (future/past sufficiency) and low rank (Cai et al., 2022; Wang et al., 2022), latent MDPs (Kwon et al., 2021; Zhou et al., 2022), learning short-memory policies (Uehara et al., 2022b), and deterministic transitions (Uehara et al., 2022a). Our definitions of $k$-MO-revealing POMDPs and distinguishable POMDPs can be seen as additional examples for tractably learnable subclasses of POMDPs under a stronger feedback ($k$-MOMDP).

More recently, Zhan et al. (2022); Chen et al. (2022a); Liu et al. (2022b); Zhong et al. (2022) study learning in a more general setting—Predictive State Representations (PSRs), which include POMDPs as a subclass. Zhan et al. (2022) show that sample-efficient learning is possible in PSRs, and Chen et al. (2022a) propose a unified condition (B-stability, which subsume revealing POMDPs and decodable POMDPs as special cases) for PSRs, and give algorithms with sharp sample complexities. Our results on $(\alpha, k)$-MO-revealing POMDPs can be viewed as an extension of the results of Chen et al. (2022a) for revealing POMDPs, adapted to the multiple observation setting.

**POMDPs with enhanced feedback**   Another line of work studies various enhanced feedback models for POMDPs (Kakade et al., 2023; Amortila et al., 2022; Li et al., 2021; Lee et al., 2023). Kakade et al. (2023) propose an interactive access model in which the algorithm can query for

samples from the conditional distributions of the Hidden Markov Models (HMMs). Shi et al. (2023) study the POMDP with partial hindsight state information, in which the agent can get access to a partial representation of the latent states. In their model, the agent immediately observes the partial information of the latent states, whereas, in our work, the agent gets the information after an episode. Closely related to our work, Lee et al. (2023) study the Hindsight Observable Markov Decision Processes (HOMDPs) as a special type of POMDPs, where the latent states are revealed to the learner in hindsight. Our feedback model can be viewed as a conceptual weakening of their model. Yet we remark that neither is strictly stronger than the other (learner can use neither one to simulate the other); see Section 3 for details. Also, in the fully observable setting, Amortila et al. (2022); Li et al. (2021) have studied feedback models similar to ours where the learner could backtrack and revisit previous states.

**Distribution testing**  Our analyses for distinguishable POMDPs build on several techniques from the distribution testing literature (Paninski, 2008; Andoni et al., 2009; Indyk et al., 2012; Ba et al., 2011; Valiant & Valiant, 2011; Goldreich & Ron, 2011; Batu et al., 2013; Acharya et al., 2015; Chan et al., 2013); see (Canonne, 2020) for a review. Notably, our OST algorithm builds on subroutines for the *closeness testing* problem, which involves determining whether two distributions over a set with $n$ elements are $\varepsilon$-close from samples. Batu et al. (2013) were the first to formally define this problem, proposing a tester with sub-linear (in $n$) sample complexity with any failure probability $\delta$. Subsequent work by Chan et al. (2013) introduced testers whose sample complexity was information-theoretically optimal for the closeness testing problem with a constant probability. The sample complexity of their tester in $\ell_1$ norm is $\Theta\left(\max\{n^{2/3}/\varepsilon^{4/3}, n^{1/2}/\varepsilon^2\}\right)$. Our OST algorithm uses an adapted version of their tester and the technique of Batu et al. (2013) to determine whether two latent states are identical with any failure probability through the multiple observations emitted from them.

## 2 PRELIMINARIES

**Notations**  For any natural number $n \in \mathbb{N}$, we use $[n]$ to represent the set $[n] = 1, 2, \ldots, n$. We use $\mathbf{I}_m$ to denote the identity matrix within $\mathbb{R}^{m \times m}$. For vectors, we denote the $\ell_p$-norm as $\|\cdot\|_p$ and $\|\cdot\|_{p \to p}$, and the expression $\|x\|_A$ represents the square root of the quadratic form $x^\top A x$. Given a set $\mathcal{S}$, we use $\Delta(\mathcal{S})$ to denote the set of all probability distributions defined on $\mathcal{S}$. For an operator $\mathbb{O}$ defined on $\mathcal{S}$ and a probability distribution $b \in \Delta(\mathcal{S})$, the notation $\mathbb{O}b : \mathcal{O} \to \mathbb{R}$ denotes the integral of $\mathbb{O}(o \mid s)$ with respect to $b(s)$, where the integration is performed over the entire set $\mathcal{S}$. For two series $\{a_n\}_{n \geq 1}$ and $\{b_n\}_{n \geq 1}$, we use $a_n \leq O(b_n)$ to mean that there exists a positive constant $C$ such that $a_n \leq C \cdot b_n$. For $\lambda \geq 0$, we use $\mathrm{Poi}(\lambda)$ to denote the Poisson distribution with parameter $\lambda$.

**POMDPs**  In this work, we study partially observable Markov decision processes (POMDPs) with a finite time horizon, denoted as $\mathcal{P}$. The POMDP can be represented by the tuple $\mathcal{P} = \left(\mathcal{S}, \mathcal{A}, H, \mathcal{O}, d_0, \{r_h\}_{h=1}^H, \{\mathbb{T}_h\}_{h=1}^H, \{\mathbb{O}_h\}_{h=1}^H\right)$, where $\mathcal{S}$ denotes the state space, $\mathcal{A}$ denotes the set of possible actions, $H \in \mathbb{N}$ represents the length of the episode, $\mathcal{O}$ represents the set of possible observations, and $d_0$ represents the initial distribution over states, which is assumed to be known. The transition kernel $\mathbb{T}_h : \mathcal{S} \times \mathcal{A} \to \mathcal{S}$ describes the probability of transitioning from one state to another state after being given a specific action at time step $h$. The reward function $r_h : \mathcal{O} \to [0, 1]$ assigns a reward to each observation in $\mathcal{O}$, and $\mathbb{O}_h : \mathcal{S} \to \Delta(\mathcal{O})$ is the observation distribution function at time step $h$. For a given state $s \in \mathcal{S}$ and observation $o \in \mathcal{O}$, $\mathbb{O}_h(o \mid s)$ represents the probability of observing $o$ while in state $s$. Note that (with known rewards and initial distribution) a POMDP can be fully described by the parameter $\theta = (\{\mathbb{T}_h\}_{h=1}^H, \{\mathbb{O}_h\}_{h=1}^H)$. We use $\tau_h := (o_{1:h}, a_{1:h}) = (o_1, a_1, \cdots, o_{h-1}, a_{h-1}, o_h, a_h)$ to denote a trajectory of observations and actions at time step $h \in [H]$. We use $S$, $A$, $O$ to denote the cardinality of $\mathcal{S}$, $\mathcal{A}$, and $\mathcal{O}$ respectively.

A policy $\pi$ is a tuple $\pi = (\pi_1, \ldots, \pi_H)$, where $\pi_h : \tau_{h-1} \times \mathcal{O} \to \Delta(\mathcal{A})$ is a mapping from histories up to step $h$ to actions. We define the value function for $\pi$ for model $\theta$ by $V_\theta(\pi) = \mathbb{E}^{\mathcal{P}}_{o_{1:H} \sim \pi}[\sum_{h=1}^H r_h(o_h)]$, namely as the expected reward received by following $\pi$. We use $V^*(\theta) = \max_\pi V_\theta(\pi)$ and $\pi^*(\theta) = \arg \max_\pi V_\theta(\pi)$ to denote the optimal value function and optimal policy for a model $\theta$. We denote the parameter of the true POMDP as $\theta^*$. We also use the shorthand $V(\pi) := V_{\theta^*}(\pi)$.

## 3 POMDPS WITH MULTIPLE OBSERVATIONS

In this section, we propose POMDPs with $k$ multiple observations ($k$-MOMDP), a new enhanced feedback model for learning POMDPs defined as follows. In the $t$-th round of interaction, the learner

1. Plays an episode in the POMDP with a policy $\pi^t$, and observes the standard trajectory feedback $\tau^t = (o_1^{t,(1)}, a_1^t, \cdots, o_H^{t,(1)}, a_H^t)$ (without observing the latent states $\{s_h^t\}_{h \in [H]}$).

2. Receives $k - 1$ additional observations $o_h^{t,(2:k)} \overset{\text{iid}}{\sim} \mathbb{O}_h(\cdot | s_h^t)$ for $h \in [H]$ *after the episode ends*.

At $k = 1$, the feedback model is the same as the standard trajectory feedback. At $k > 1$, the $k - 1$ additional observations cannot affect the trajectory $\tau^t$ but can reveal more information about the past encountered latent states, which could be beneficial for learning (choosing the policy for the next round). We remark that such a "replay" ability has also been explored in several recent works, such as Lee et al. (2023) who assume that the learner could know the visited states after each iteration, and Amortila et al. (2022); Li et al. (2021) who assume that the learner could reset to any visited states then continue to take actions to generate a trajectory.

We consider a general setting where the value of $k$ in $k$-MOMDP can be determined by the learner. Consequently, for a fair comparison of the sample complexities, we take into account all observations (both the trajectory and the $(k - 1)$ additional observations) when counting the number of samples, so that each round of interaction counts as $kH$ observations/samples.

**Relationship with the hindsight observable setting**  Closely related to $k$-MOMDP, Lee et al. (2023) propose the *hindsight observable* setting, another feedback model for learning POMDPs in which the learner directly observes the true latent states $\{s_h^t\}_{h \in [H]}$ after the $t$-th episode. In terms of their relationship, neither feedback model is stronger than (can simulate) the other in a strict sense, when learning from bandit feedback: Conditioned on the $k - 1$ additional observations, the true latent state could still be random; Given the true latent state, the learner in general, does not know the emission distribution to simulate additional samples. However, our multiple observation setting is conceptually "weaker" (making learning harder) than the hindsight observatbility setting, as the true latent state is exactly revealed in the hindsight observable setting but only "approximately" revealed in our setting through the noisy channel of multiple observations in hindsight.

A natural first question about the $k$-MOMDP feedback model is that whether it fully resolves the hardness of learning in POMDPs (for example, making any tabular POMDP learnable with polynomially many samples). The following result shows that the answer is negative.

**Proposition 1** (Existence of POMDP not polynomially learnable under $k$-MO feedback for any finite $k$). *For any $H, A \geq 2$, there exists a POMDP with $H$ steps, $A$ actions, and $S = O = 2$ (non-revealing combination lock) that cannot be solved with $o(A^{H-1})$ samples with high probability under $k$-MOMDP feedback for any $k \geq 1$.*

Proposition 1 shows that some structural assumption on the POMDP is necessary for it to be sample-efficiently learnable in $k$-MOMDP setting (proof can be found in Appendix B.1), which we investigate in the sequel. This is in contrast to the hindsight observable setting (Lee et al., 2023) where any tabular POMDP can be learned with polynomially many samples, and suggests that $k$-MOMDP as an enhanced feedback model is in a sense more relaxed.

## 4  $k$-MO-REVEALING POMDPS

We now introduce the class of $k$-MO-revealing POMDPs, and show that they are sample-efficiently learnable under $k$-MOMDP feedback.

### 4.1  DEFINITION

To introduce this class, we begin by noticing that learning POMDPs under the $k$-MOMDP feedback can be recast as learning an *augmented* POMDP under standard trajectory feedback. Indeed, we can simply combine the observations during the episode and the hindsight into an augmented observation $\{o_h^{(1:k)}\}_{h \in [H]}$ which belongs to $\mathcal{O}^k = \{o^{(1:k)} : o^{(i)} \in \mathcal{O}\}$. The policy class that the learner optimizes over in this setting is a *restricted* policy class (denoted as $\Pi_{\text{singleobs}}$) that is only allowed to depend on the first entry $o_h^{(1)}$ instead of the full augmented observation $o_h^{(1:k)}$.

We now present the definition of a $k$-MO revealing POMDP, which simply requries its augmented POMDP under the $k$-MOMDP feedback is (single-step) revealing.  For any matrix $\mathbb{O} = \{\mathbb{O}(o|s)\}_{o,s \in \mathcal{O} \times \mathcal{S}} \in \mathbb{R}^{\mathcal{O} \times \mathcal{S}}$ and any $k \geq 1$, let $\mathbb{O}^{\otimes k} \in \mathbb{R}^{\mathcal{O}^k \times \mathcal{S}}$ denote the column-wise $k$ self-tensor of $\mathbb{O}$, given by $\mathbb{O}^{\otimes k}(o_{1:k}|s) = \prod_{i=1}^k \mathbb{O}(o_i|s)$.

**Definition 2** (MO-revealing POMDP). *For any $k \geq 1$ and $\alpha \in (0, 1]$, a POMDP is said to be $(\alpha, k)$-MO-revealing if its augmented POMDP under the $k$-MOMDP feedback is $\alpha$-revealing. In other words, a POMDP is $k$-MO-revealing if for all $h \in [H]$, the matrix $\mathbb{O}_h^{\otimes k}$ has a left inverse $\mathbb{O}_h^{\otimes k+} \in \mathbb{R}^{S \times O^k}$ (i.e. $\mathbb{O}_h^{\otimes k+} \mathbb{O}_h^{\otimes k} = \mathbf{I}_S$) such that*

$$\left\| \mathbb{O}_h^{\otimes k+} \right\|_{1 \rightarrow 1} \leq \alpha^{-1}.$$

Above, we allow *any* left inverse of $\mathbb{O}_h^{\otimes k}$ and use the matrix $(1 \rightarrow 1)$ norm to measure the revealing constant following Chen et al. (2023), which allows a tight characterization of the sample complexity.

As a basic property, we show that $(\alpha, k)$-MO-revealing POMDPs are strictly larger subclasses of POMDPs as $k$ increases. The proof can be found in Appendix B.2.

**Proposition 3** (Relationship between $(\alpha, k)$-MO-revealing POMDPs). *For all $\alpha \in (0, 1]$ and $k \geq 1$, any $(\alpha, k)$-MO-revealing POMDP is also an $(\alpha, k + 1)$-MO-revealing POMDP. Conversely, for all $k \geq 2$, there exists a POMDP that is $(\alpha, k + 1)$-MO-revealing for some $\alpha > 0$ but not $(\alpha', k)$-MO-revealing for any $\alpha' > 0$.*

Proposition 3 shows that $(\alpha, k)$-MO-revealing POMDPs are systematic relaxations of the standard $\alpha$-revealing POMDPs (Jin et al., 2020c; Liu et al., 2022a; Chen et al., 2023), which corresponds to the special case of $(\alpha, k)$-MO-revealing with $k = 1$. Intuitively, such relaxations are also natural, as the $k$-multiple observation setting makes it *easier* for the observations to reveal information about the latent state in any POMDP. We remark in passing that the containment in Proposition 3 is strict.

### 4.2 ALGORITHM AND GUARANTEE

In this section, we first introduce the $k$-OMLE algorithm and then provide the theoretical guarantee of $k$-OMLE for the low-rank POMDPs.

**Algorithm: $k$-Optimistic Maximum Likelihood Estimation ($k$-OMLE)**  Here, we provide a brief introduction to Algorithm $k$-OMLE. The algorithm is an adaptation of the OMLE algorithm (Liu et al., 2022a; Zhan et al., 2022; Chen et al., 2022a; Liu et al., 2022b) into the $k$-MOMDP feedback setting. As noted before, we can cast the problem of learning under $k$-MOMDP feedback as learning in an augmented POMDP with the restricted policy class $\Pi_{\mathsf{singleobs}}$. Then, the $k$-OMLE algorithm is simply the OMLE algorithm applied in this problem.

Concretely, each iteration $t \in [T]$ of the $k$-OMLE algorithm consists of two primary steps:

1. The learner executes exploration policies $\{\pi_{h,\exp}^t\}_{0 \leq h \leq H-1}$, where each $\pi_{h,\exp}^t$ is defined via the $\circ_{h-1}$ notation as follows: It follows $\pi^t$ for the first $h - 1$ steps, then takes the uniform action $\mathrm{Unif}(\mathcal{A})$, and then takes arbitrary actions (for example using $\mathrm{Unif}(\mathcal{A})$ afterwards (Line 5). All collected trajectories are then incorporated into $\mathcal{D}$ (Line 6).

2. The learner constructs a confidence set $\Theta^t$ within the model class $\Theta$, which is a super level set of the log-likelihood of all trajectories within the dataset $\mathcal{D}$ (Line 7). The policy $\pi^k$ is then selected as the greedy policy with respect to the most optimistic model within $\Theta^k$ (Line 3).

**Theoretical guarantee**  Our guarantee for $k$-OMLE requires the POMDP to satisfy the $k$-MO-revealing condition and an additional guarantee on its rank, similar to existing work on learning POMDPs (Wang et al., 2022; Chen et al., 2022a; Liu et al., 2022b). For simplicity of the presentation, here we consider the case of POMDPs with low-rank latent transitions (which includes tabular POMDPs as a special case); our results directly hold in the more general case where $d$ is the *PSR rank* of the problem (Chen et al., 2022a; Liu et al., 2022b; Zhong et al., 2022).

**Definition 4** (Low-rank POMDP (Zhan et al., 2022; Chen et al., 2022a)). *A POMDP $\mathcal{P}$ is called low-rank POMDP with rank $d$ if its transition kernel $\mathbb{T}_h : \mathcal{S} \times \mathcal{A} \rightarrow \mathcal{S}$ admits a low-rank decomposition of dimension $d$, i.e. there exists two mappings $\mu_h^* : \mathcal{S} \rightarrow \mathbb{R}^d$, and $\phi_h^* : \mathcal{S} \times \mathcal{A} \rightarrow \mathbb{R}^d$ such that $\mathbb{T}_h(s' \mid s, a) = \mu_h^*(s')^\top \phi_h^*(s, a)$.*

We also make the standard realizability assumption that the model class contains the true POMDP: $\theta^* \in \Theta$ (but otherwise does not require that the mappings $\{\mu_h^*, \phi_h^*\}_h$ are known).

We state the theoretical guarantee for $k$-OMLE on low-rank POMDPs. The proof follows directly by adapting the analysis of Chen et al. (2022a) into the augmented POMDP (see Appendix C.1).

---

**Algorithm 1** $k$-Optimistic Maximum Likelihood Estimation ($k$-OMLE)

---

**Input:** Model class $\Theta$, parameter $\beta > 0$, and $k \in \mathbb{N}$.
1: **Initialize:** $\Theta_1 = \Theta$, $\mathcal{D} = \varnothing$.
2: **for** iteration $t = 1, \cdots, T$ **do**
3:     Set $\theta^t, \pi^t = \arg\max_{\theta \in \Theta^t, \pi} V_\theta(\pi)$.
4:     **for** $h = 0, \cdots, H - 1$ **do**
5:         Set exploration policy $\pi^t_{h,\exp} := \pi^t \circ_{h-1} \mathrm{Unif}(\mathcal{A})$.
6:         Execute $\pi^t_{h,\exp}$ to collect a $k$-observation trajectory $\tau^{t,h}_k$, and add $(\pi^t_{h,\exp}, \tau^{t,h}_k)$ to $\mathcal{D}$, where $\tau^{t,h}_k = \left( o_1^{t,(1:k)}, a_1, \ldots, o_H^{t,(1:k)}, a_H^{t,(1:k)} \right)$ as in Section 3.
7:     Update confidence set
$$\Theta^{t+1} = \left\{ \widehat{\theta} \in \Theta : \sum_{(\pi, \tau_t) \in \mathcal{D}} \log \mathbb{P}^\pi_{\widehat{\theta}}(\tau_t) \geqslant \max_{\theta \in \Theta} \sum_{(\pi, \tau_t) \in \mathcal{D}} \log \mathbb{P}^\pi_\theta(\tau_t) - \beta \right\}.$$
8: **Return** $\pi^T$.

---

**Theorem 5** (Results of $k$-OMLE for $(\alpha, k)$-MO-revealing low-rank POMDPs). *Suppose the true model $\theta^*$ is a low-rank POMDP with rank $d$, is realizable ($\theta^* \in \Theta$), and every $\theta \in \Theta$ is $(\alpha, k)$-MO-revealing. Then choosing $\beta = \mathcal{O}(\log(\mathcal{N}_\Theta/\delta))$, with probability at least $1 - \delta$, Algorithm 1 outputs a policy $\pi^T$ such that $V^* - V(\pi^T) \leqslant \varepsilon$ within*

$$N = THk = \widetilde{\mathcal{O}}\left( \mathrm{poly}(H) k d A \log \mathcal{N}_\Theta / \left( \alpha^2 \varepsilon^2 \right) \right)$$

*samples. Above, $\mathcal{N}_\Theta$ is the optimistic covering number of $\Theta$ defined in Appendix C.*

We also state a result of tabular $(\alpha, k)$-revealing POMDPs. Note that any tabular POMDP is also a low-rank POMDP with rank $d = SA$, hence Theorem 5 applies; however the result below achieves a slightly better rate (by using the fact that the PSR rank is at most $S$).

**Theorem 6** (Results of $k$-OMLE for $(\alpha, k)$-MO-revealing tabular POMDPs). *Suppose $\theta^*$ is $(\alpha, k)$-MO-revealing and $\Theta$ consists of all tabular $(\alpha, k)$-MO-revealing POMDPs. Then, choosing $\beta = \mathcal{O}(H \left( S^2 A + SO \right) + \log(1/\delta))$, then with probability at least $1 - \delta$, Algorithm 1 outputs a policy $\pi^T$ such that $V^* - V(\pi^T) \leqslant \varepsilon$ within the followinng number of samples:*

$$\widetilde{\mathcal{O}}\left( \mathrm{poly}(H) k S A (S^2 A + SO) / \left( \alpha^2 \varepsilon^2 \right) \right).$$

We remark that the rate asserted in Theorem 5 & 6 also hold for the Explorative Estimation-To-Decisions (Explorative E2D) (Foster et al., 2021) and the Model-based Optimistic Posterior Sampling (Agarwal & Zhang, 2022) algorithms (with an additional low-rank requirement on every $\theta \in \Theta$ for Explorative E2D), building upon the unified analysis framework of Chen et al. (2022b;a). See Appendix C.1 for details.

## 5 DISTINGUISHABLE POMDPs

Given $k$-MO-revealing POMDPs as a first example of sample-efficiently learnable POMDPs under $k$-MOMDP feedback, it is of interest to understand the minimal structure required for learning under this feedback. In this section, we investigate a natural proposal—distinguishable POMDPs, and study its relationship with $k$-MO-revealing POMDPs as well as sample-efficient learning algorithms.

### 5.1 DEFINITION

The definition of distinguishable POMDPs is motivated by the simple observation that, if there exist two states $s_i, s_j \in \mathcal{S}$ that admit *exactly the same emission distributions* (i.e. $\mathbb{O}_h e_i = \mathbb{O}_h e_j \in \Delta(\mathcal{O})$), then the two states are not distinguishable under $k$-MOMDP feedback no matter how large $k$ is. Our formal definition makes this quantitiative, requiring any two states to admit $\alpha$-different emission distributions in the $\ell_1$ (total variation) distance.

**Definition 7** (Distinguishable POMDP). *For any $\alpha \in (0, 1]$, a POMDP is said to be $\alpha$-distinguishable if for all $h \in [H]$ (where $e_i \in \mathbb{R}^\mathcal{S}$ denotes the $i$-th standard basis vector),*

$$\min_{i \neq j \in \mathcal{S}} \|\mathbb{O}_h (e_i - e_j)\|_1 \geq \alpha.$$

*Qualitatively, we say a POMDP is distinguishable if it is $\alpha$-distinguishable for some $\alpha > 0$.*

Notably, distinguishability only requires the emission matrix $\mathbb{O}_h$ to have *distinct* columns. This is much weaker than the (single-step) revealing condition (Jin et al., 2020a; Liu et al., 2022a; Chen et al., 2022a; 2023) which requires $\mathbb{O}_h$ to have *linearly independent* columns. In other words, in a distinguishable POMDP, different latent states may not be probabilistically identifiable from a single observation as in a revealing POMDP; however, this does not preclude the possibility that we can identify the latent state with $k > 1$ observations.

Also, we focus on the natural case of tabular POMDPs (i.e. $S$, $A$, $O$ are finite) when considering distinguishable POMDPs[1], and leave the question extending or modifying the definition to infinitely many states/observations to future work.

## 5.2 Relationship with $k$-MO-revealing POMDPs

We now study the relationship between distinguishable POMDPs and $k$-MO-revealing POMDPs. Formal statement and proof of the following results see Appendix D.1 and Appendix D.2.

We begin by showing that any $k$-MO revealing POMDP is necessarily a distinguishable POMDP. This is not surprising, as distinguishability is a necessary condition for $k$-MO-revealing—if distinguishability is violated, then there exists two states with identical emission distributions and thus identical emission distributions with $k$ iid observations for any $k \geq 1$, necessarily violating $k$-MO-revealing.

**Proposition 8** ($k$-MO revealing POMDPs $\subset$ Distinguishable POMDPs). *For any $\alpha \in (0, 1]$, $k \geq 1$, any $(\alpha, k)$-revealing tabular POMDP is a distinguishable POMDP.*

Perhaps more surprisingly, we show that the reverse containment is also true in a sense if we allow $k$ to be large—Any $\alpha$-distinguishable POMDP is also a $k$-MO revealing POMDP for a suitable large $k$ depending on $(S, O, \alpha)$, and revealing constant $\Theta(1)$.

**Theorem 9** (Distinguishable $\subset$ MO-revealing with large $k$). *There exists an absolute constant $C > 0$ such that any $\alpha$-distinguishable POMDP is also $(1/2, k)$-MO-revealing for any $k \geq C\sqrt{O}\log(SO)/\alpha^2$.*

**Proof by embedding a distribution test**    The proof of Theorem 9 works by showing that, for any distinguishable POMDP, the $k$-observation emission matrix $\mathbb{O}_h^{\otimes k}$ admits a well-conditioned left inverse with a suitably large $k$. The construction of such a left inverse borrows techniques from distribution testing literature, where we *embed* an identity test (Batu et al., 2013; Chan et al., 2013) with $k$ observations into a $\mathcal{S} \times \mathcal{O}^k$ matrix, with each column consisting of indicators of the test result. The required condition for this matrix to be well-conditioned (and thus $\mathbb{O}_h^{\otimes k}$ admitting a left inverse) is that $k$ is large enough—precisely $k \geq \widetilde{\mathcal{O}}(\sqrt{O}/\alpha^2)$ (given by known results in identity testing (Chan et al., 2013))—such that the test succeeds with high probability.

**Sample-efficient learning by reduction to $k$-MO-revealing case**    Theorem 9 implies that, since any $\alpha$-distinguishable POMDP is also a $(1/2, k)$-MO-revealing POMDP with $k = \mathcal{O}(\sqrt{O}\log(SO)/\alpha^2)$, it can be efficiently learned by the $k$-OMLE algorithm, with number of samples

$$\widetilde{\mathcal{O}}\left(\mathrm{poly}(H)SA\sqrt{O}(S^2A + SO)/\left(\alpha^2\varepsilon^2\right)\right) \tag{1}$$

given by Theorem 6. This shows that any distinguishable POMDP is sample-efficiently learnable under $k$-MOMDP feedback by choosing a proper $k$.

## 5.3 Sharper Algorithm: OST

We now introduce a more efficient algorithm—Optimism with State Testing (OST; Algorithm 2)—for learning distinguishable POMDPs under $k$-MOMDP feedback.

Recall that in a distinguishable POMDP, different latent states have $\alpha$-separated emission distributions, and we can observe $k$ observations per state. The main idea in OST is to use *closeness testing* algorithms (Chan et al., 2013; Batu et al., 2013) to determine whether any two $k$-fold observations are from the same state. As long as all pairwise tests return correct results, we can perform a *clustering* to recover a "pseudo" state label that is guaranteed to be the correct latent states up to a permutation.

---

[1]When $S$ is infinite (e.g. if the state space $\mathcal{S}$ is continuous), requiring any two emission distributions to differ by $\alpha$ in $\ell_1$ distance may be an unnatural requirement, as near-by states could yield similar emission probabilities in typical scenarios.

---

**Algorithm 2** Optimism with State Testing (OST)

---

**Input:** POMDP planner POP, parameters $\beta_1, \beta_2 > 0$ and $k \in \mathbb{N}$.

1: **Initialize:** Emission and transition models $O_1, T_1$, initial pseudo state space $[n_1^h] = \emptyset$ (i.e. $n_1^h = 0$) and initial visitation counts $n_h^1(s) = n_h^1(s,a) = 0$ for all $s \in \widetilde{\mathcal{S}}_1, a \in \mathcal{A}$ and $h \in [H]$.

2: **for** iteration $t = 1, \cdots, T$ **do**

3:     **for** all $(s,a) \in [S] \times \mathcal{A}$ **do**

4:         Set $b_t(s,a) = \min\left\{\sqrt{\beta_1/n_t(s,a)}, 2H\right\}$ and $b_t(s) = \min\left\{\sqrt{\beta_2/n_t(s)}, 2\right\}$ as the exploration bonus.

5:         Set $\widehat{r}_h^t(s,a) = \min\{1, \bar{r}_h^t(s,a) + Hb_t(s) + b_t(s,a)\}$, where $\bar{r}_t$ is defined in (2).

6:     Update $\pi^t = POP(\widehat{T}_t, \widehat{O}_t, \widehat{r}_t)$.

7:     Execute $\pi^t$ to collect a $k$-observation trajectory $\tau_k^t$, where $\tau_k^t = \left(o_1^{t,(1:k)}, a_1, \ldots, o_H^{t,(1:k)}, a_H\right)$.

8:     Call ASSIGN_PSEUDO_STATES (Algorithm 3) to obtain pseudo states $\{\widetilde{s}_h^t\}_{h \in [H]}$.

9:     Set $n_h^{t+1}(s) = \sum_{l \in [t], h \in [H]} 1\{\widetilde{s}_h^l = s\}$ for all $(h,s) \in [H] \times [S]$.

10:    Set $n_h^{t+1}(s,a) = \sum_{l \in [t], h \in [H]} 1\{\widetilde{s}_h^l = s \wedge a_h^l = a\}$ for all $(h,s,a) \in [H] \times [S] \times \mathcal{A}$.

11:    Update $\widehat{T}_{t+1}$ and $\widehat{O}_{t+1}$ by (3).

12: **return** $\pi^t$.

---

**Algorithm 3** Pseudo state assignment via closeness testing (ASSIGN_PSEUDO_STATES)

---

1: **for** $h \in [H]$ **do**

2:    assigned $= 0$.

3:    **for** $\widetilde{s} \in [n_h^t]$ **do**

4:       **if** closeness_test$(o_h^{t,(1:k)}, o_h^{t',(1:k)}) = $ **accept** (Algorithm 4) for all $t' \in [\widetilde{s}_h^{t'} = \widetilde{s}]$ **then**

5:          Set $\widetilde{s}_h^t = \widetilde{s}$, assigned $= 1$, $n_h^{t+1} = n_h^t$, **break**

6:       **if** assigned $= 0$ **then**

7:          Set $\widetilde{s}_h^t = n_h^t + 1$, $n_h^{t+1} = n_h^t + 1$.

---

Given the pseudo states, we can then adapt the techniques from the hindsight observable setting (Lee et al., 2023) to accurately estimate the model of the POMDP and learn a near-optimal policy.

**Algorithm description** We first define a planning oracle (Lee et al., 2023; Jin et al., 2020b), which serves as an abstraction of the optimal planning procedure that maps any POMDP $(T, O, r)$ to an optimal policy of it.

**Definition 10** (POMDP Optimal Planner). *The POMDP planner POP takes as input a transition function $T := \{T_h\}_{h=1}^H$, an emission function $O := \{O_h\}_{h=1}^H$, and a reward function $r : \mathcal{S} \times \mathcal{A} \to [0,1]$ and returns a policy $\pi = \mathrm{POP}(T, O, r)$ to maximize the value function under the POMDP with latent transitions $\{T_h\}_{h=1}^H$, emissions $\{O_h\}_{h=1}^H$, and reward function $r$.*

OST operates over $T$ rounds, beginning with arbitrary initial $\widehat{T}_1$ and $\widehat{O}_1$ of the model. We set the initial pseudo state space as an empty set. Then, at each iteration $t$, OST calculates reward bonuses $b_t(s,a)$ and $b_t(s)$ to capture the uncertainty of $\widehat{T}_t$ and $\widehat{O}_t$, quantified by the number of visits to each latent state in the pseudo state space (Line 4). The bonuses defined in (2) are added to the following empirical reward estimates (Line 5). We then call POP to calculate the policy for the current iteration and deploy it to obtain a $k$-observation trajectory from the $k$-MOMDP feedback (Line 6-7).

$$\bar{r}_h^t(s,a) = \sum_{o \in \mathcal{O}} \sum_{\ell \in [t]} \frac{r(o)\mathbf{1}\left\{s_h^\ell = s, o_h^\ell = o\right\}}{\min\{1, n_h^t(s)\}}. \tag{2}$$

We next employ closeness testing and clustering to assign pseudo states $(\widetilde{s}_1^t, \ldots, \widetilde{s}_H^t)$ to the trajectory $\tau_k^t$ (Line 8) using Algorithm 3. For each $k$-observation $o_h^{t,(1:k)}$ generated in the new iteration, we perform a closeness test with all past $\{o_h^{t',(1:k)}\}_{t' > t}$ to check if they belong to the same pseudo state: Two states are the same state if their $k$ observations pass closeness testing, different if they fail it. Using the test results, we perform a simple "clustering" step: If $o_h^{t,(1:k)}$ passes the closeness test against all $\{o_h^{t',(1:k)}\}$ who has been assigned as pseudo state $\widetilde{s}$, then we assign $\widetilde{s}$ to $o_h^{t,(1:k)}$. If the state is not assigned after all tests, then that indicates the encountered latent state has not been encountered

---

**Algorithm 4** Closeness Testing closeness_test($\{o^{(i)}\}_{i\in[k]}, \{\widetilde{o}^{(i)}\}_{i\in[k]}$)

---

**Input:** $[o^{[i]}]_{i\in[k]}, [\widetilde{o}^{[i]}]_{i\in[k]}$
1: Sample $N_1, \cdots, N_M \sim \text{Poi}(k/M)$, where $M = \mathcal{O}(\log(1/\delta))$.
2: **return fail** if $N_1 + \cdots + N_M > k$.
3: **for** $j \in [M]$ **do**
4: $\quad B_j = \{N_1 + \cdots + N_{j-1} + 1, \cdots, N_1 + \cdots + N_j\}$.
5: $\quad N_o^{(j)} = \sum_{i\in B_j} \mathbf{1}\{o_i = o\}, \widetilde{N}_o^{(j)} = \sum_{i\in B_j} \mathbf{1}\{\widetilde{o}_i = o\}$.
6: $\quad Z^{(j)} = \mathbf{1}\{\sum_{o\in\mathcal{O}}((N_o^{(j)} - \widetilde{N}_o^{(j)})^2 - N_o^{(j)} - \widetilde{N}_o^{(j)})/(N_o^{(j)} + \widetilde{N}_o^{(j)}) \leq \sqrt{3N_j}\}$.
7: **return accept** if $\sum_{j\in[M]} z^{(j)} \geq M/2$, **else reject**

---

before (not in the current pseudo state space $[n_h^t]$), in which case we assign $o_h^{t,(1:k)}$ with a new pseudo state $n_h^t + 1$, which enlarges the pseudo state space to $[n_h^{t+1}] = [n_h^t + 1]$.

Our particular closeness testing algorithm (Algorithm 4) adapts the test and analysis in Chan et al. (2013) and makes certain modifications, such as repeating a test with a Poisson number of samples $\log(1/\delta)$ times to reduce the failure probability from a $\Theta(1)$ constant to $\delta$, as well as imposing a hard upper limit $k$ on the total sample size (so that the test can be implemented under the $k$-MOMDP feedback), instead of a Poisson number of samples which is unbounded.

Finally, using the assigned pseudo states, we update the visitation counts of each (pseudo) state $s$ and state-action $(s, a)$ (Line 9-10). Then we update the pseudo latent models $(\widehat{T}_{t+1}, \widehat{O}_{t+1}) = (\{\widehat{T}_h^{t+1}\}_{h=1}^H, \{\widehat{O}_h^{t+1}\}_{h=1}^H)$ using empirical estimates based on the previous data (Line 11):

$$\widehat{T}_h^{t+1}(s' \mid s, a) = \sum_{\ell\in[t]} \frac{\mathbf{1}\{s_h^\ell = s, s_{h+1}^\ell = s'\}}{\min\{1, n_h^{t+1}(s,a)\}}, \quad \widehat{O}_h^{t+1}(o \mid s) = \sum_{\ell\in[t]} \frac{\mathbf{1}\{s_h^\ell = s, o_h^\ell = o\}}{\min\{1, n_h^{t+1}(s)\}}. \tag{3}$$

**Theoretical guarantee** We now present the main guarantee for OST for learning distinguishable POMDPs. The proof can be found in Appendix D.3. Further, OST is computationally efficient given the planning oracle $POP$, as all other operations in Algorithm 2 takes $\text{poly}(H, S, A, O, T, k)$ time.

**Theorem 11** (Learnining distinguishable POMDPs by OST). *For any $\alpha$-distinguishable POMDP, choosing $\beta_1 = \mathcal{O}(H^3 \log(OSAHK/\delta)), \beta_2 = \mathcal{O}(O\log(OSKH/\delta))$ and $k = \widetilde{\mathcal{O}}((\sqrt{O}/\alpha^2 + O^{2/3}/\alpha^{4/3}))$, with probability at least $1 - \delta$, the output policy of Algorithm 2 is $\varepsilon$-optimal after the following number of samples:*

$$\widetilde{\mathcal{O}}\left(\text{poly}(H) \cdot \left(\frac{SO}{\varepsilon^2} + \frac{SAk}{\varepsilon^2}\right)\right) = \widetilde{\mathcal{O}}\left(\text{poly}(H) \cdot \left(\frac{SO}{\varepsilon^2} + \frac{SA\sqrt{O}}{\varepsilon^2\alpha^2} + \frac{SAO^{2/3}}{\varepsilon^2\alpha^{4/3}}\right)\right).$$

The proof of Theorem 11 builds on high-probability correctness guarantees of closeness_test, which enables us to adapt the algorithm and analysis of the hindsight observable setting Lee et al. (2023) if all tests return the correct results (so that pseudo states coincide with the true latent states up to a permutation). Compared to the rate obtained by $k$-OMLE (Eq. (1)), Theorem 11 achieves a better sample complexity (all three terms above are smaller the $S^2AO^{1.5}/(\alpha^2\varepsilon^2)$ term therein, ignoring $H$ factors). Technically, this is enabled by the *explicit* closeness tests built into OST combined with a sharp analysis of learning tabular POMDPs with observed latent states, rather than the implicit identity tests used in the reduction approach (Theorem 9) with the $k$-OMLE algorithm.

## 6 CONCLUSION

In this paper, we investigated $k$-Multiple Observations MDP ($k$-MOMDP), a new enhanced feedback model that allows efficient learning in broader classes of Partially Observable Markov Decision Processes (POMDPs) than under the standard feedback model. We introduced two new classes of POMDPs—$k$-MO-revealing POMDPs and distinguishable POMDPs and designed sample-efficient algorithms for learning in these POMDPs under $k$-MOMDP feedback. Overall, our results shed light on the broader question of when POMDPs can be efficiently learnable from the lens of enhanced feedbacks. We believe our work opens up many directions for future work, such as lower bounds for the sample complexities, identifying alternative efficiently learnable classes of POMDPs under $k$-MOMDP feedback, generalizing distinguishable POMDPs to the function approximation setting, or developing more computationally efficient algorithms.

## 7 ACKNOWLEDGEMENTS

Mengdi Wang acknowledges the support by NSF grants CPS-2312093, DMS-1953686, IIS-2107304, CMMI1653435, ONR grant 1006977 and C3.AI.

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

## A    TECHNICAL LEMMAS

**Lemma A.1.** Suppose $Y_s \in \{0,1\}^{\mathcal{O}^k}$ for all $s \in \mathcal{S}$, and the locations of the $1's$ are disjoint within the rows $s \in \mathcal{S}$. The matrix $\mathbb{Y}$ is defined by

$$\mathbb{Y} := \begin{bmatrix} Y_1^\top \\ \vdots \\ Y_S^\top \end{bmatrix} \in \mathbb{R}^{\mathcal{S} \times \mathcal{O}^k}.$$

Then we have

$$\|\mathbb{Y}\|_{1 \to 1} = 1.$$

*Proof of Lemma A.1.* Let's analyze the action of $\mathbb{Y}$ on an arbitrary non-zero vector $\mathbf{x} \in \mathbb{R}^{\mathcal{O}^k}$. Since each column of $\mathbb{Y}$ has at most one non-zero element, which is 1, the action of $\mathbb{Y}$ on $\mathbf{x}$ is:

$$\mathbb{Y}\mathbf{x} = \begin{bmatrix} \sum_{i=1}^{\mathcal{O}^k} Y_{1,i} x_i \\ \vdots \\ \sum_{i=1}^{\mathcal{O}^k} Y_{S,i} x_i \end{bmatrix} = \begin{bmatrix} x_{j_1} \\ \vdots \\ x_{j_S} \end{bmatrix}$$

Here, $x_{j_s}$ is the element of $\mathbf{x}$ corresponding to the non-zero entry in row $s$ of $\mathbb{Y}$. Then, we have:

$$\|\mathbb{Y}\| \, \mathbf{x} = \sum_{s=1}^{S} |x_{j_s}| \le \sum_{i=1}^{\mathcal{O}^k} |x_i| = \|x\|.$$

It follows that $\frac{\|\mathbb{Y}\mathbf{x}\|_1}{\|\mathbf{x}\|_1} \le 1$ for any non-zero $\mathbf{x} \in \mathbb{R}^{\mathcal{O}^k}$, so $\|\mathbb{Y}\|_{1\to 1} \le 1$.

Now let's find a non-zero vector $\mathbf{x}$ for which $\frac{\|\mathbb{Y}\mathbf{x}\|_1}{\|\mathbf{x}\|_1} = 1$. Let $\mathbf{x}$ be the vector with all elements equal to 1, i.e., $x_i = 1$ for all $i$. Then, the action of $\mathbb{Y}$ on $\mathbf{x}$ is:

$$\mathbb{Y}\mathbf{x} = \begin{bmatrix} 1 \\ \vdots \\ 1 \end{bmatrix}.$$

This gives us $\|\mathbb{Y}\|_{1\to 1} \ge 1$.

Combining the two inequalities, we finish the proof of Lemma A.1.    $\square$

**Lemma A.2.** Suppose $E$ is a matrix satisfies that $\|E\|_{1\to 1} < 1$, then $\mathbf{I} + E$ is invertible.

*Proof.* To prove this lemma, we will show that $\mathbf{I} + E$ has no eigenvalue equal to zero. If there 0 is an eigenvalue and there exists $x \ne 0$, s.t.

$$(\mathbf{I} + E)\vec{x} = 0,$$

which means

$$\|x\|_1 = \|Ex\|_1.$$

This implies that $\|E\|_{1\to 1} \ge 1$, which contradicts with the fact that $\|E\|_{1\to 1} < 1$. Hence $\mathbf{I} + E$ must be invertible.    $\square$

## B    PROOFS FOR SECTION 3

### B.1    PROOF OF PROPOSITION 1

*Proof.* Inspired by the bad case in Liu et al. (2022a), we construct a combination lock to prove the proposition, which is defined as follows:

Consider two states, labeled as a "good state" ($s_g$) and a "bad state" ($s_b$), and two observations, $o_g$ and $o_b$. For the initial $H - 1$ steps, the emission matrices are

$$\begin{pmatrix} 1/2 & 1/2 \\ 1/2 & 1/2 \end{pmatrix}.$$

while at step $H$, the emission matrix becomes

$$\begin{pmatrix} 1 & 0 \\ 0 & 1 \end{pmatrix}.$$

This implies that no information is learned at each step $h \in [H - 1]$, but at step $H$, the current latent state is always directly observed.

In our model, there are $A$ different actions with the initial state set as $s_g$. The transitions are defined as follows: For every $h \in [H - 1]$, one action is labeled "good", while the others are "bad". If the agent is in the "good" state and makes the "good" action, it remains in the "good" state. In all other scenarios, the agent transitions to the "bad" state. The good action is randomly selected from $\mathcal{A}$ for each $h \in [H - 1]$. The episode concludes immediately after $o_H$ is obtained.

All observations during the initial $H - 1$ steps get a reward of 0. At step $H$, observation $o_g$ produces a reward of 1, while observation $o_b$ yields 0. Therefore, the agent receives a reward of 1 solely if it reaches the state $s_g$ at step $H$, i.e., if the optimal action is taken at every step.

Assume we attempt to learn this POMDP using an algorithm $\mathcal{X}$, where we are given $T$ episodes, with $k$-multiple observations, to interact with the POMDP. Regardless of the selection of $k$, no information can be learned at step $h \in [H - 1]$, making the best strategy a random guess of the optimal action sequence. More specifically, the probability that $\mathcal{X}$ incorrectly guesses the optimal sequence, given that we have $T$ guesses, is

$$\binom{A^{H-1} - 1}{T} / \binom{A^{H-1}}{T} = (A^{H-1} - T)/A^{H-1}.$$

For $T \leq A^{H-1}/10$, this value is at least $9/10$. Therefore, with a minimum probability of 0.9, the agent learns nothing except that the chosen action sequences are incorrect, and the best policy it can produce is to randomly select from the remaining action sequences, which is less efficient than $(1/2)$-optimal. This concludes our proof. □

### B.2 PROOF OF PROPOSITION 3

*Proof.* First, we prove the existence of an $(\alpha, k + 1)$-MO-revealing POMDP $\mathcal{P}$ that is not an $(\alpha, k)$-MO-revealing POMDP. To this end, it suffices to consider a POMDP with a single step ($H = 1$) with emission matrix $\mathbb{O} \equiv \mathbb{O}_1 \in \mathbb{R}^{O \times S}$.

We consider the following POMDP with $H = 1$, $O = 2$ and $S = k + 2$. We denote $\mathcal{O} = \{o_1, o_2\}$ and $\mathcal{S} = \{s_1, \cdots, s_{k+2}\}$. The probabilities $\mathbb{O}(o_1 \mid s_i)$ and $\mathbb{O}(o_2 \mid s_i)$ for $i \in [k + 2]$ are denoted as $u_i$ and $v_i$ respectively, with $\{v_1, \ldots, v_{k+2}\} \subset (0, 1)$ being distinct. It should be noted that $u_i + v_i = 1$ for any $i$.

We first consider the rank of $\mathbb{O}^{\otimes k} \in \mathbb{R}^{2^k \times (k+2)}$. Note that $\mathbb{O}^{\otimes k}(o^{1:k}|s)$ only depends on the number of $o_1$'s and $o_2$'s within $o^{1:k}$, which only has $k + 1$ possibilities ($o_1^k, o_1^{k-1}o_2, \cdots, o_2^k$). Therefore, $\mathbb{O}^k$ only has at most $(k + 1)$ distinct rows, and thus $\text{rank}(\mathbb{O}^{\otimes k}) \leq k + 1$. Since $k + 1 < \min\{2^k, k + 2\}$ for all $k \geq 2$, $\mathbb{O}^{\otimes k}$ is rank-deficient, and thus the constructed POMDP with emission matrix $\mathbb{O}$ is not an $(\alpha', k)$-MO-revealing POMDP for any $\alpha' > 0$.

Next, we consider the rank of $\mathbb{O}^{\otimes k+1}$. Using similar arguments as above, $\mathbb{O}^{\otimes k+1}$ has $(k + 2)$ distinct rows, and thus its rank equals the rank of the corresponding $(k + 2) \times (k + 2)$ submatrix:

$$\begin{bmatrix} u_1^{k+1} & u_2^{k+1} & \cdots & u_{k+2}^{k+1} \\ u_1^k v_1 & u_2^k v_2 & \cdots & u_{k+2}^k v_{k+2} \\ \vdots & \vdots & \ddots & \vdots \\ v_1^{k+1} & v_2^{k+1} & \cdots & v_{k+2}^{k+1} \end{bmatrix}.$$

By rescaling each column $i$ with $1/u_i^{k+1}$, the resulting matrrix is a Vandermonde matrix generated by distinct values $\{v_i/u_i\}_{i\in[k+2]}$, and thus has full rank (Boyd & Vandenberghe, 2018, Exercise 6.18). This implies that $\mathbb{O}^{\otimes k+1}$ is full-rank and thus admits a finite left inverse (for example its pseudo inverse) $(\mathbb{O}^{\otimes k+1})^+$ with finite $(1 \to 1)$ norm. This shows the constructed POMDP is $(\alpha, k+1)$-MO-revealing with $\alpha = \left\|(\mathbb{O}^{\otimes k+1})^+\right\|_{1\to 1}^{-1} > 0$.

Now we prove that any $(\alpha, k)$-MO-revealing POMDP $\mathcal{P}$ is also an $(\alpha, k+1)$-MO-revealing POMDP. Let $\mathcal{P}$ be an $(\alpha, k)$-revealing POMDP. Fix any $h \in [H]$ and let $\mathbb{O} \equiv \mathbb{O}_h$ for shorthand. Let $(a_{ij})_{ij}$ represent the $\mathbb{O}^{\otimes k}$ matrix of $\mathcal{P}$, where $i \in \mathcal{O}^k$ and $j \in \mathcal{S}$. Let $(b_{ij})_{ij}$ denote the $\mathbb{O}^{\otimes k+}$ matrix of $\mathcal{P}$, where $i \in \mathcal{S}$ and $j \in \mathcal{O}^k$.

By the definition of $\mathbb{O}^{\otimes k+}$, we have the following equations:

$$\sum_{o_1 \cdots o_k \in \mathcal{O}^k} b_{s,o_1 \cdots o_k} a_{o_1 \cdots o_k, s'} = \mathbf{1}\left\{s = s'\right\}, \quad \forall s, s' \in \mathcal{S}. \tag{4}$$

Let $(\bar{a}_{ij})_{ij}$ represent the $\mathbb{O}^{\otimes k+1}$ matrix of $\mathcal{P}$, where $i \in \mathcal{O}^{k+1}$ and $j \in \mathcal{S}$. Note that we have $\bar{a}_{o_1 \cdots o_{k+1}, s} = a_{o_1 \cdots o_k, s} \mathbb{O}(o_{k+1} \mid s)$.

Now we construct $\mathbb{O}^{\otimes k+1+} = (\bar{b}_{ij})_{ij}$ as $\bar{b}_{s, o_1 \cdots o_{k+1}} := b_{s, o_1 \cdots o_k}$. For all $s, s' \in \mathcal{S}$, we have:

$$\sum_{o_1 \cdots o_{k+1} \in \mathcal{O}^{k+1}} \bar{b}_{s, o_1 \cdots o_{k+1}} \bar{a}_{o_1 \cdots o_{k+1}, s'} = \sum_{o_1 \cdots o_{k+1} \in \mathcal{O}^{k+1}} b_{s, o_1 \cdots o_k} a_{o_1 \cdots o_k, s'} \mathbb{O}(o_{k+1} \mid s')$$

$$= \sum_{o_1 \cdots o_k \in \mathcal{O}^k} b_{s, o_1 \cdots o_k} a_{o_1 \cdots o_k, s'}$$

$$= \mathbf{1}\left\{s = s'\right\},$$

where the last equation follows from (4). This shows that the constructed $(\mathbb{O}^{\otimes k+1})^+$ is indeed a left inverse of $\mathbb{O}^{\otimes k+1}$. Further, for any vector $v \in \mathbb{R}^{\mathcal{O}^{k+1}}$, we have

$$\left\|(\mathbb{O}^{\otimes k+1})^+ v\right\|_1 = \sum_{s \in \mathcal{S}} \sum_{o_{1:k+1}} \left|\bar{b}_{s, o_1 \cdots o_{k+1}} v_{o_1 \cdots o_{k+1}}\right| = \sum_{s \in \mathcal{S}} \sum_{o_{1:k+1}} \left|b_{s, o_1 \cdots o_k} v_{o_1 \cdots o_{k+1}}\right|$$

$$\leq \sum_{o_{k+1}} \left\|(\mathbb{O}^{\otimes k})^+ v_{:, o_{k+1}}\right\|_1 \leq \alpha^{-1} \sum_{o_{k+1}} \left\|v_{:, o_{k+1}}\right\|_1 = \alpha^{-1} \left\|v\right\|_1.$$

This shows that $\left\|(\mathbb{O}^{\otimes k+1})^+\right\|_{1\to 1} \leq \alpha^{-1}$. Since the above holds for any $h \in [H]$, we have $\mathcal{P}$ is an $(\alpha, k+1)$-revealing POMDP.

## C  PROOFS FOR SECTION 4

### C.1  PROOF OF THEOREM 5 AND THEOREM 6

First, we define the optimistic cover and optimistic covering number for any model class $\Theta$ and $\rho > 0$. The definition is taken from Chen et al. (2022b).

**Lemma C.1** (Optimistic cover and optimistic covering number (Chen et al., 2022b))**.** Suppose that there is a context space $\mathcal{X}$. An optimistic $\rho$-cover of $\Theta$ is a tuple $\left(\widetilde{\mathbb{P}}, \Theta_0\right)$, where $\Theta_0 \subset \Theta$ is a finite set, $\widetilde{\mathbb{P}} = \left\{\widetilde{\mathbb{P}}_{\theta_0}(\cdot) \in \mathbb{R}_{\geqslant 0}^{\mathcal{T}^H}\right\}_{\theta_0 \in \Theta_0, \pi \in \Pi}$ specifies a optimistic likelihood function for each $\theta \in \Theta_0$, such that:

1. For $\theta \in \Theta$, there exists a $\theta_0 \in \Theta_0$ satisfying: for all $\tau \in \mathcal{T}^H$ and $\pi$, it holds that $\widetilde{\mathbb{P}}_{\theta_0}^\pi(\tau) \geqslant \mathbb{P}_\theta^\pi(\tau)$;

2. For $\theta \in \Theta_0, \max_\pi \left\|\mathbb{P}_\theta^\pi(\tau_H = \cdot) - \widetilde{\mathbb{P}}_\theta^\pi(\tau_H = \cdot)\right\|_1 \leqslant \rho^2$.

The optimistic covering number $\mathcal{N}_\Theta(\rho)$ is defined as the minimal cardinality of $\Theta_0$ such that there exists $\widetilde{\mathbb{P}}$ such that $\left(\widetilde{\mathbb{P}}, \mathcal{M}_0\right)$ is an optimistic $\rho$-cover of $\Theta$.

Remind that we consider the $k$-MOMDP as an augmented POMDP and note that we are going to find the optimized policy in $\Pi_{\text{singleobs}}$, which only depends on the single immediate observation $o_h^{(1)}$.

Our $k$-OMLE algorithm can be seen as an adaptation of Algorithm OMLE in Chen et al. (2022a) to the augmented POMDP with policy class $\Pi_{\text{singleobs}}$.

Chen et al. (2022a) showed that OMLE achieves the following estimation bound for low-rank POMDPs.

**Theorem C.2** (Theorem 9 in Chen et al. (2022a)). Choosing $\beta = \mathcal{O}(\log(\mathcal{N}_\Theta/\delta))$, then with probability at least $1 - \delta$, Algorithm OMLE outputs a policy $\pi^T$ such that $V_* - V_{\theta^*}\left(\pi^T\right) \leqslant \varepsilon$, after

$$\widetilde{\mathcal{O}}\left(\text{poly}(H)d_{PSR}A\log\mathcal{N}_\Theta/\left(\alpha^2\varepsilon^2\right)\right)$$

samples, where they considered POMDPs as Predictive State Representations (PSRs), and $d_{PSR}$ is the PSR rank. In our setting, the $\text{poly}(H)$ is $H^5$ (accounting for the additional $H^2$ factor introduced by reward scaling and $H$ factor by counting the number of samples instead of episodes). For low-rank POMDP, $d_{PSR} \leq d$, where $d$ is the rank of the decomposition of the transition kernel.

In the proof of Theorem 9 in Chen et al. (2022a), the use of the policy class is based on the fact that the policy $\pi^t$ at each iteration is chosen to be the optimal policy within the model confidence set $\Theta^t$ for that round, specifically $\pi^t = \arg\max_{\theta\in\Theta^t,\pi} V_\theta(\pi)$. By replacing the policy class with $\Pi_{\text{res}}$, we still maintain this property, ensuring that the chosen policy remains optimal within the updated model confidence set. Therefore, the replacement is valid and does not affect the optimality of the selected policies throughout the algorithm. In $k$-OMLE, we can select the model class as an $O(\varepsilon/k)$ covering of the original model class (with a single observation), which will induce an $O(\varepsilon)$-covering of the augmented POMDP class by the $k$-fold product structure of the augmented emission matrix. Therefore, since the covering number only depends logarithmically on the precision parameter, the $\varepsilon$-covering number for the augmented POMDP class will not change when omitting the logarithmic term. Hence, by invoking Theorem C.2 and the above argument, we obtain the convergence rate stated in Theorem 5.

For tabular POMDPs with $S$ states, the PSR rank becomes $S$. Additionally, Liu et al. (2022a) showed that $\log\mathcal{N}_\Theta(\rho) \leq \mathcal{O}(H(S^2A + SO)\log(HSOA/\rho))$. By utilizing this result and repeating the discussion about the covering number as above, the covering number is also $\widetilde{O}\left(H(S^2A + SO)\right)$. Therefore, we can derive the convergence rate for the tabular case. $\qquad\square$

**Extension to Explorative E2D and MOPS** The above augmentation (considering $k$-MOMDP as an augmented POMDP and searching in $\Pi_{\text{singleobs}}$) can also be applied to extend Theorem 10 in Chen et al. (2022a). The theorem is achieved by Algorithm Explorative Estimation-to-Decisions (Explorative E2D). Furthermore, it can be extended to Theorem F.6 in Chen et al. (2022a), which is achieved by Model-based optimistic posterior sampling (MOPS). The OMLE, Explorative E2D, and MOPS extension to $k$-MOMDPs share the same sample complexity rates.

MOPS and E2D require slightly stronger conditions compared to OMLE. While OMLE only necessitates $\theta^*$ to possess the low PSR rank structure, E2D requires every model within $\Theta$ to exhibit the same low-rank structure. All three algorithms require every model within $\Theta$ to be k-MO-revealing, not just $\theta^*$.

# D  PROOFS FOR SECTION 5

## D.1  PROOF OF PROPOSITION 8

*Proof.* We utilize proof by contradiction to establish the validity of this problem. Suppose we have a tabular POMDP, denoted by $\mathcal{P}$, that is not distinguishable. Hence there exists $i, j \in \mathcal{S}$, such that

$$\|\mathbb{O}_h(e_i - e_j)\|_1 = 0,$$

which means that there must exist two different states, $s_1$ and $s_2$ belonging to $\mathcal{S}$, such that they share the same emission kernels. As a result, the columns associated with $s_1$ and $s_2$ will be identical. Consequently, the $k$-fold tensor power of the observation space $\mathbb{O}^{\otimes k}$ for $\mathcal{P}$ will be a rank-deficient matrix, implying that it lacks a left inverse. This leads us to conclude that $\mathcal{P}$ cannot be a revealing POMDP. This contradiction substantiates our original proposition, hence completing the proof. $\quad\square$

## D.2 Proof of Theorem 9

*Proof.* Consider any $\alpha$-distinguishable POMDP and any fixed $h \in [H]$.

**Step 1.** By lemma D.3, we construct tests $Z_s = \{Z_s(o_{1:k})\}_{o_{1:k} \in \mathcal{O}^k}$ for each $s \in \mathcal{S}$, such that $Z_s \leq 1/2$ with probability at least $1 - \delta$ under $\mathbb{O}_h(\cdot|s)$, and $Z_s \geq 1$ with probability at least $1 - \delta$ under $\mathbb{O}_h(\cdot|s')$ for any $s' \neq s$.

**Step 2.** For every $s \in \mathcal{S}$, define "identity test for latent state $s$":

$$Y_s(o_{1:k}) = \mathbf{1}\left\{Z_s(o_{1:k}) = \min_{s' \in \mathcal{S}} Z_{s'}(o_{1:k})\right\} \in \{0, 1\},$$

with an arbitrary tie-breaking rule for the min (such as in lexicographic order). Understand $Y_s \in \mathbb{R}^{\mathcal{O}^k}$ as a vector. Define matrix

$$\mathbb{Y}_h := \begin{bmatrix} Y_1^\top \\ \vdots \\ Y_S^\top \end{bmatrix} \in \mathbb{R}^{\mathcal{S} \times \mathcal{O}^k}.$$

By step 1, we have

$$\mathbb{Y}_h \mathbb{O}_h^{\otimes k+} = \mathbf{I}_\mathcal{S} + E,$$

where the matrix $E \in \mathbb{R}^{\mathcal{S} \times \mathcal{S}}$ satisfies $|E_{ij}| \leq S\delta$. We pick $\delta = 1/(2S^2)$ (which requires $k \geq \sqrt{O}/\alpha^2 \log 1/\delta$). Further, notice that each $Y_s \in \{0, 1\}^{\mathcal{O}^k}$, and the **locations of the 1's are disjoint** within the rows $s \in \mathcal{S}$. By Lemma A.1 we have

$$\|\mathbb{Y}_h\|_{1 \to 1} = 1.$$

**Step 3.** Notice that (where $\{e_s^\top\}_{s \in \mathcal{S}}$ are rows of $E$)

$$\|E\|_{1 \to 1} = \max_{\|x\|_1 = 1} \sum_{s \in \mathcal{S}} |e_s^\top x| \leq \sum_{s \in \mathcal{S}} \|e_s\|_\infty \leq S^2 \delta \leq 1/2.$$

By Lemma A.2 we know that $\mathbf{I}_\mathcal{S} + E$ is invertible. Further, we have

$$\left\|(\mathbf{I}_\mathcal{S} + E)^{-1}\right\|_{1 \to 1} = \left\|\mathbf{I}_\mathcal{S} + \sum_{k=1}^\infty (-1)^k E^k\right\|_{1 \to 1} \leq 1 + \sum_{k=1}^\infty \|E\|_{1 \to 1}^k = \frac{1}{1 - \|E\|_{1 \to 1}} \leq 2.$$

Finally, define the matrix

$$\mathbb{O}_h^{\otimes k+} := (\mathbf{I}_\mathcal{S} + E)^{-1} \mathbb{Y}_h \in \mathbb{R}^{\mathcal{S} \times \mathcal{O}^k}.$$

We have $\mathbb{O}_h^{\otimes k+} \mathbb{O}_h^{\otimes k} = (\mathbf{I}_\mathcal{S} + E)^{-1} \mathbb{Y}_h \mathbb{O}_h^{\otimes k} = (\mathbf{I}_\mathcal{S} + E)^{-1}(\mathbf{I}_\mathcal{S} + E) = \mathbf{I}_\mathcal{S}$. Further,

$$\left\|\mathbb{O}_h^{\otimes k+}\right\|_{1 \to 1} \leq \left\|(\mathbf{I}_\mathcal{S} + E)^{-1}\right\|_{1 \to 1} \cdot \|\mathbb{Y}_h\|_{1 \to 1} \leq 2.$$

This completes the proof. $\qquad \square$

## D.3 Proof of Theorem 11

*Proof.* First, we introduce the Hindsight Observable Markov Decision Processes (HOMDPs), POMDPs where the latent states are revealed to the learner in hindsight.

**HOMDP (Lee et al., 2023)** There are two phases in the HOMDP: train time and test time. During train time, at any given round $t \in [T]$, the learner produces a history-dependent policy $\pi^t$ which is deployed in the partially observable environment as if the learner is interacting with a standard POMDP. Once the $t$ th episode is completed, the latent states $s_{1:H}^t$ are revealed to the learner in hindsight, hence the terminology hindsight observability. Lee et al. (2023) showed that HOP-B achieves the following estimation bound for HOMDPs.

**Theorem D.1** (Theorem 4.2 in Lee et al. (2023))**.** Let $\mathcal{M}$ be a HOMDP model with $S$ latent states and $O$ observations. With probability at least $1 - \delta$, HOP-B outputs a sequence of policies $\pi^1, \dots, \pi^T$ such that

$$\text{Reg}(T) = \widetilde{\mathcal{O}}\left(H^{\frac{5}{2}} \sqrt{(SO + SA)T}\right).$$

Adapted to modified setting where each latent state yields $K$ observations, by adapting their proof (using all $K$ observations collectively to estimate the emission matrices), we have

$$\text{Reg}(T) = \widetilde{\mathcal{O}}\left(H^{\frac{5}{2}}\sqrt{(SO/K + SA)T}\right). \tag{5}$$

Our proof is a reduction from the result (5) combined with results for closeness testing to ensure that states are correct (up to permutation). Suppose in algorithm 2, the pseudo-states are indeed true states (up to permutation). Then, our algorithm 2 achieves the sample complexity

$$TkH = \widetilde{\mathcal{O}}\left(H^6 \cdot \left(\frac{SO}{\varepsilon^2} + \frac{SA\sqrt{O}}{\varepsilon^2\alpha^2} + \frac{SAO^{2/3}}{\varepsilon^2\alpha^{4/3}}\right)\right).$$

Now we explain our proof, our Algorithm 2 are different from HOP-B in two points:

1. HOP-B works for HOMDP, where the exact information of the pseudo-states can be immediately known. In $k$-MOMDP, we cannot determine the exact state even when we can distinguish all the states. Therefore, Algorithm 2 reduces to the HOP-B algorithm up to permute.

2. Since we cannot know the exact pseudo-states, we couldn't assume the reward on the pseudo state space $\mathcal{X}$ is known for each $s \in \mathcal{X}$. We can only estimate the reward towards the estimated emission kernel.

$$\bar{r}_h^t(s,a) = \sum_{o\in\mathcal{O}}\sum_{\ell\in[t]}\frac{r_h(o)\mathbf{1}\left\{s_h^\ell = s, o_h^\ell = o\right\}}{n_h^t(s)} = \sum_{o\in\mathcal{O}}\widehat{\mathbb{O}}_t(o\mid s)r_h(o),$$

where $\widehat{\mathbb{O}}_t$ is the estimated emission kernel in the $t$-th iteration. This leads to an extra error between the estimated reward function $\bar{r}$ and true reward function $r_h(s,a) = \sum_{o\in\mathcal{O}}\mathbb{O}(o\mid s)r_h(o)$.

Since we assume the reward function $r$ can be bounded by 1, the error between the estimated reward function $\bar{r}$ and true reward function $r(s,a)$ can be bounded as:

$$\bar{r}_h(s,a) - r_h(s,a) = \sum_{o\in\mathcal{O}}(\widehat{\mathbb{O}}_t(o\mid s) - \mathbb{O}(o\mid s))r_h(o)$$
$$\leq \sum_{o\in\mathcal{O}}\|\widehat{\mathbb{O}}_t(o\mid s) - \mathbb{O}(o\mid s)\|.$$

which can be bounded by $\sqrt{(O\log(SOTH/\delta))/(n_t(s))}$ with probability $1-\delta$ as showed in Lee et al. (2023). Hence we can additionally handle the reward estimation, however, this will not result in a change of the rate, as we can just choose a larger constant in the exploration bonus for states in their HOP-B algorithm (line4, $\beta_2$) to ensure optimism still holds.

The previous theorem requires the pseudo-states to be true. To ensure this requires the guarantee of closeness testing, which we give here, we will prove it in Section D.4. We state that with a high probability, we could identify the pseudo-states (up to permutation), which means that after an iteration, we could know whether the states visited in this iteration were visited before. We use the closeness testing algorithm to test whether two observation sequences were generated from the same state.

**Lemma D.2** (Closeness testing guarantee). When $k = \mathcal{O}((\sqrt{O}/\alpha^2 + O^{2/3}/\alpha^{4/3})\log 1/\delta)$, with probability $1-\delta$ the following holds: Throughout the execution of Algorithm 2, we have that there exists a permutation $\pi : \mathcal{S} \to \mathcal{S}$, such that pseudo-states are up to permutation of true states (we could identify each state).

After identifying the pseudo-states (up to permutation), we can think that we have information about the state after each iteration, as we said in Section 3.

On the success event of closeness testing, states are indeed correct. Therefore we can invoke (5) to obtain the sample complexity

$$TkH = \widetilde{\mathcal{O}}\left(H^6 \cdot \left(\frac{SO}{\varepsilon^2} + \frac{SAk}{\varepsilon^2}\right)\right) = \widetilde{\mathcal{O}}\left(H^6 \cdot \left(\frac{SO}{\varepsilon^2} + \frac{SA\sqrt{O}}{\varepsilon^2\alpha^2} + \frac{SAO^{2/3}}{\varepsilon^2\alpha^{4/3}}\right)\right)$$

by taking $k = \mathcal{O}((\sqrt{O}\alpha^2 + O^{2/3}/\alpha^{4/3})\log 1/\delta)$ to the bound of Theorem D.1, which finish the proof. $\qquad\square$

## D.4 CLOSENESS TESTING

In this section, we prove the theoretical guarantee for closeness testing.

*Proof of Lemma D.2.* Let's assume $X \sim \text{Poi}(\lambda)$. We have tail bound for $X$: for any $x > 0$,

$$\mathbb{P}(X > \lambda + x) \leq e^{x - (\lambda + x)\ln(1 + \frac{x}{\lambda})}.$$

Employing this tail bound, we conclude that Algorithm 4 will not return a fail with a probability of $1 - \mathcal{O}(\delta)$. Our subsequent analysis is contingent upon this event.

Based on Proposition 3 in Chan et al. (2013), given $k = \mathcal{O}((\sqrt{O}\alpha^2 + O^{2/3}/\alpha^{4/3})\log 1/\delta)$, we can infer that for any $j \in [M]$: $Z^{(j)} = 1$ with a probability of at least $2/3$ if $o_h^{t,(1:k)}$ and $o_h^{t',(1:k)}$ generated from the same state, $Z^{(j)} = 0$ with a probability of at least $2/3$ if they are produced by different states. Utilizing standard repeating techniques, we find that with $\mathcal{O}\left(\log\frac{1}{\delta}\right)$ iterations, we can attain an error probability of at most $\delta$.

Thus, we've established that if $o_h^{t,(1:k)}$ and $o_h^{t',(1:k)}$ are generated from the same state, Algorithm 4 will return an accept with a probability of $1 - \delta$. Conversely, if they are generated from different states, the algorithm 4 will return a reject with a probability of $1 - \delta$. $\qquad\square$

---

**Algorithm 5** Closeness Testing 2 $\mathsf{closeness\_test2}([o^{[i]}]_{i\in[k]})$

---

**Input:** $[o^{[i]}]_{i\in[k]}$
1: Sample $N_1, \cdots, N_M \sim \text{Poi}(k/M)$, where $M = \mathcal{O}(\log(1/\delta))$.
2: **return fail if** $N_1 + \cdots + N_M > k$.
3: **for** $j \in [M]$ **do**
4:      $B_j = \{N_1 + \cdots + N_{j-1} + 1, \cdots, N_1 + \cdots + N_j\}$.
5:      $N_o^{(j)} = \sum_{i \in B_j} \mathbf{1}\{o_i = o\}$.
6:      $A^{(j)} = \{o : o \geq \alpha/(50O)\}$.
7:      $C^{(j)} = \sum_{o \in A^{(j)}} ((N_o^{(j)} - N_j q_o)^2 - N_o^{(j)})/N_j q_o)$.
8:      $z^{(j)} = \mathbf{1}\{C^{(j)} \leq N_j \alpha^2/10\}$.
9: **return accept if** $\sum_{j \in [M]} z^{(j)} \geq M/2$, **else reject**

---

**Lemma D.3.** Suppose $\mathcal{P}$ is an $\alpha$-distinguishable POMDP, then we can construct tests $Z_s = \{Z_s(o_{1:k})\}_{o_{1:k} \in \mathcal{O}^k}$ for each $s \in \mathcal{S}$, such that $Z_s = 0$ with probability at least $1 - \delta$ under $\mathbb{O}_h(\cdot|s)$, and $Z_s = 1$ with probability at least $1 - \delta$ under $\mathbb{O}_h(\cdot|s')$ for any $s' \neq s$.

*Proof.* We construct $Z_s$ by using the closeness testing technique.

First, we give to consider a problem: Given samples from an unknown distribution $p$, is it possible to distinguish whether $p$ equal to $\mathbb{O}$ versus $p$ being $\alpha$-far from every $\mathbb{O}$?

Chan et al. (2013) proposes an algorithm to achieve its lower bound $\sqrt{O}/\alpha^2$. We improve their algorithm by repeating $\log(1/\delta)$ times to attain an error probability of at most $\delta$. We denote $q_o = \mathbb{O}(o \mid s)$. The algorithm is listed in Algorithm 5.

Finally, we apply the repeating technique to Theorem 2 in Chan et al. (2013). Then we set $k = \sqrt{O}/\alpha^2 \log 1/\delta$ and set $Z_s(o_{1:k}) = \mathbf{1}\{\mathsf{closeness\_test2}(o_{1:k}) = \text{accept}\}$. Hence we can identify whether an observation sequence $o_{1:k}$ is generated from state $s$ with probability $1 - \delta$. Therefore, we complete the proof of Lemma D.3. $\qquad\square$

