# OpenReview forum: "Sample-Efficient Learning of POMDPs with Multiple Observations In Hindsight"
_ICLR.cc/2024/Conference — ICLR 2024 poster_

### Official Review · Reviewer_jqb5 · 2023-10-27

**Soundness:** 3 good
**Presentation:** 3 good
**Contribution:** 2 fair
**Rating:** 6
**Confidence:** 4

**Summary:**

This work studies the problem of sample-efficient learning in POMDP. Especially, a new feedback model is discussed, i.e., multiple observations in hindsight, where instead of one observation in canonical POMDP setting, after one trajectory, multiple extra observations can be obtained. This feedback model extends the canonical single-observation feedback model in POMDP and is suitable to capture applications such as game-playing where replaying or loading is enabled.

Based on this new feedback model, one new revealing condition is presented, i.e., multi-observation revealing, which is extended from canonical revealing conditions in POMDP. Based on this condition, the k-OMLE design is first proposed, which is demonstrated to be sample efficient. Then, a second class of POMDP, defined as distinguishable POMDP, is also introduced, which relies on the intuition that different states should have distinct observation generation. The relationship between multi-observation revealing POMDP and distinguishable POMDP is established. Furthermore, leveraging the techniques from closeness testing and POMDP with hindsight observation, a new design, OST, is proposed, which is also demonstrated to be sample efficient.

**Strengths:**

- The POMDP problem has received growing interest in recent years and this work is a valuable investigation following the line of identifying learnable subclass of POMDP, which makes reasonable contribution.

- The overall motivation is also clear, i.e., having a class of POMDP with stronger feedback than single observation while still weaker than directly revealing the true state. Having multiple observations is also a valid and practical consideration in my view as indeed replaying or loading is often allowed in settings such as game-playing.

- The paper is organized well and the overall flow is quite clear. Although being technically heavy, the key points and main intuitions are explained well.

**Weaknesses:**

- The major concern that I have is that this work mainly rely on techniques from previous works. Especially, k-OMLE, as the authors stated, is a different instantiation of the OMLE algorithm, and the proof is also extended rather straightforwardly. While I do believe using closeness testing and the notion of distinguishable POMDP are interesting, the OST design is then about returning to the POMDP design with exact hindsight observability.

**Questions:**

- First, it would be really helpful if the authors can illustrate more on the technical innovation of this work. Especially, I would love to know whether the authors believe the main contribution of this work is to identify new subclasses of POMDP while the leveraged techniques are mostly same as previous works.

- Also, in the design of OST, as stated in Theorem 11, the value of \alpha is required to determine the value of k, which seems to be a rather stringent condition. I would love to hear about the authors' opinion on this.

---

> ### Author Response · Authors · 2023-11-18
> **Response to Reviewer jqb5**
>
> We thank the reviewer for the valuable feedback on our paper. We respond to the specific questions as follows.
>
> >... the technical innovation of this work. Especially, I would love to know whether the authors believe the main contribution of this work is to identify new subclasses of POMDP while the leveraged techniques are mostly same as previous works.
>
> At a high level, we agree with the reviewer that identifying the two new subclasses (k-MO-revealing POMDPs, distinguishable POMDPs) is a main contribution, and constructing the algorithms build on existing works once they are identified.
>
> However, besides just identifying the two classes, we also studied their relationship in Proposition 8 and Theorem 9. In particular, Theorem 9 which shows that any $\alpha$-distinguishable POMDP is $k$-MO-revealing for a certain $k$, is shown by “embedding” distribution tests to construct left inverses of the $k$-fold emission matrix. Such an argument is new to the RL literature and we believe is technically interesting as well.
>
> > In the design of OST, as stated in Theorem 11, the value of \alpha is required to determine the value of k, which seems to be a rather stringent condition. I would love to hear about the authors' opinions on this.
>
> We agree that, algorithmically, if all we know for the POMDP is it’s $\alpha$ distinguishable, then we need to choose $k$ based on $\alpha$. In practice, if $\alpha$ is known to have a lower bound $\alpha_0$, then we can choose $k$ based on $\alpha_0$.
>
> On the other hand, if we do not have knowledge about $\alpha$, then we can always choose any fixed $k$ to learn, and we can still achieve good sample complexity if the POMDP is $k$-MO revealing.

---

> > ### Comment · Reviewer_q33U · 2023-11-22
> >
> > I acknowledge that I have read the authors' response and other reviews.

---

### Official Review · Reviewer_q33U · 2023-11-01

**Soundness:** 4 excellent
**Presentation:** 4 excellent
**Contribution:** 3 good
**Rating:** 6
**Confidence:** 3

**Summary:**

The paper introduces two new subclasses of POMDP, multi-observation revealing POMDPs and distinguishable POMDPs. The paper shows a connection between the two classes of POMDPs and shows that sample-efficient learning in both classes is possible.

**Strengths:**

The work builds on the ongoing effort to establish classes of POMDPs for which sample efficient learning is possible. The new classes of POMDPs are formally and rigorously defined, and their introduction is backed by practical motivation. Bounds on sample efficiency are established and proved, with some proofs using less common techniques of distribution testing embedded in the algorithm. I have read and I believe I understood the proofs, although I cannot swear that the proofs are correct.

**Weaknesses:**

It is not clear to me whether MO revealing POMDPs are novel --- see my question below.

It is not clear to me from the paper how hindsight observations are related/required. It seems that MO revealing POMDPs are defined independently of hindsight observability. Definition 2 does not require or that the observations are obtained in the hindsight.

The significant results of the work seems to be only applicable to tabular POMDPs, which only becomes clear deep into the paper. I believe that should be stated early.

Minor: the bibliography is not properly capitalized. Abbreviations (POMDP, PAC etc.) and names should be capitalized.

**Questions:**

1. What is the difference between k-MO revealing POMDPs in this work and MO revealing POMDPs in https://proceedings.mlr.press/v202/chen23ae.html, Definition 1?

2. How does obtaining multiple observations in **hindsight** affect the learnability/sample efficiency? Why the same sample efficiency cannot be achieved with the observations obtained online?

3. Shouldn't learnability for alpha-distinguishable POMPDs be established in a PAC setting rather than absolutely?

---

> ### Author Response · Authors · 2023-11-18
> **Response to Reviewer q33U**
>
> We thank the reviewer for the valuable feedback on our paper. We respond to the specific questions as follows.
>
> > It is not clear to me from the paper how hindsight observations are related/required. It seems that MO revealing POMDPs are defined independently of hindsight observability. Definition 2 does not require or that the observations are obtained in the hindsight.
>
> Indeed, Definition 2 is merely a property of the POMDP itself (revealing from $k$ observations), and does not require specification of when these $k$ observations are received by the learner. However, this setting is still related to hindsight observability, as they are both feedback models that enhance the standard POMDP feedback (observations and rewards only), and our later algorithm OST builds on techniques from the hindsight observability setting.
>
> > The significant results of the work seems to be only applicable to tabular POMDPs, which only becomes clear deep into the paper. I believe that should be stated early.
>
> Thank you for the suggestion, the results for distinguishable POMDPs and the OST algorithm are indeed only for tabular POMDPs. We have highlighted that in the abstract and contribution list in our revision.
>
> > What is the difference between k-MO revealing POMDPs in this work and MO revealing POMDPs in https://proceedings.mlr.press/v202/chen23ae.html, Definition 1?
>
> The difference between k-MO revealing POMDPs and the $m$-step revealing POMDPs e.g. in Chen et al. (2023) is **multi-observation** revealing vs. **multi-step** revealing:
> *  $k$-MO revealing require $k$ repeated observations at each time step to reveal some information about the current latent state,
> * $m$-step revealing requires observations and actions at $m$ consecutive steps (i.e $(o_h, a_h, \dots, o_{h+m-1})$) to reveal some information about the current latent state $s_h$.
>
> > How does obtaining multiple observations in **hindsight** affect the learnability/sample efficiency? Why the same sample efficiency cannot be achieved with the observations obtained online?
>
> We chose this setting merely to avoid a potential confusion about *policies*---Restricting the main trajectory to still have 1 observation restricts the policy itself to still be a standard POMDP policy, and prevents it from utilizing more observations to do better. We’d like to keep the policy class to be still the standard POMDP policy class, and only use the $k-1$ observations to help the learning within this class, hence we chose this setting.
>
> > Shouldn't learnability for alpha-distinguishable POMPDs be established in a PAC setting rather than absolutely?
>
> We’d like to clarify what the reviewer meant by "absolutely"? Our Theorem 11 is indeed a PAC result, where the algorithm returns a policy that is an $\epsilon$ near-optimal policy with probability at least $1-\delta$.
>
> > The bibliography is not properly capitalized. Abbreviations (POMDP, PAC etc.) and names should be capitalized.
>
> Thanks for the suggestion. We have updated the bibliography accordingly in our revision.

---

### Official Review · Reviewer_se9m · 2023-11-01

**Soundness:** 2 fair
**Presentation:** 2 fair
**Contribution:** 3 good
**Rating:** 6
**Confidence:** 3

**Summary:**

This paper studied how to achieve sample-efficient learning in POMDPs, which have been known to be exponentially hard in the worst case. How to identify efficient feedback model and provide efficient learning algorithms for the challenging POMDP problem has been an important problem in the community. This paper studied the settings with additional observations generated by the latent unknown states at the end of the episode. With these additional observations and the additional assumptions, i.e., definition 2 and definition 7, this paper proposed efficient learning algorithms with performance guarantees.

**Strengths:**

1. This paper studied the important and challenging POMDP problem, and showed two new subclasses where efficient learning is possible.

2. This paper provided efficient learning algorithms for the new subclasses with performance guarantees.

**Weaknesses:**

1. The model and assumptions are not well-justified.

2. The solution seems to be a simple extension of the existing results for weakly-revealing settings.

**Questions:**

1. Could you provide one or more practical examples for the model with multiple observations?

2. Could you explain the practical meaning of the assumptions, e.g., definition 2 and definition 7?

3. Could you give a clearer explanation for the differences between your results and those for weakly-revealing settings, since it seems the algorithm developments and performance analyses are quite similar?

---

> ### Author Response · Authors · 2023-11-18
> **Response to Reviewer se9m**
>
> We thank the reviewer for the valuable feedback on our paper. We respond to the specific questions as follows.
>
> > Could you provide one or more practical examples for the model with multiple observations?
>
> One example is in gaming problems (cf. the bottom paragraph on Page 1), where a player can load (reset to) a previously saved game state, and re-play from there. By doing that the player can make additional observations (such as results of a battle), but not observe the latent states (such as the true status/properties of the characters).
>
> Another example is a medical imaging problem, where we usually take multiple images per time step to measure the latent state (true status of the patient) that cannot be directly observed, and the problem is sequential (need to repeat this along the treatment process).
>
> > Could you explain the practical meaning of the assumptions, e.g., definition 2 and definition 7?
>
> In a more practical sense, Definition 2 ($k$-MO revealing) means that the collection of $k$ iid observations can reveal some information about the latent state, whereas Definition 7 (distinguishable) means that different latent states have different emission distributions.
>
> In the above game example, Definition 2 means that by playing the same battle $K$ times we can gain information about the true game state, whereas Definition 7 means that different game states must yield different (distributions of) battle results. We remark that Definition 2 implies Definition 7 (Proposition 8) and we show the converse is also true in a certain sense (Theorem 9).
>
>
> > Could you give a clearer explanation for the differences between your results and those for weakly-revealing settings, since it seems the algorithm developments and performance analyses are quite similar?
>
> Our results consist of two settings, k-MO-revealing POMDPs, and distinguishable POMDPs.
> * For k-MO-revealing POMDPs, our algorithm indeed builds on existing work on weakly-revealing POMDPs such as the OMLE algorithm.
> * For distinguishable POMDPs, our algorithm is quite different from weakly-revealing settings, as it builds on techniques from distribution testing and hindsight observability, and the algorithm is very different from OMLE. Further, for distinguishable POMDPs, OST gives a sharper result than a direct reduction to k-MO-revealing POMDPs (cf. the paragraph after Theorem 11).

---

> > ### Comment · Reviewer_se9m · 2023-11-22
> >
> > I acknowledge that I have read the authors' response and other reviews. I am not very convinced by the medical image example, e.g., what would the episode and initial state distribution $d_0$ mean in such an example? However, I agree with the strengths pointed out by other reviewers. I have increased my rate to a 6.

---

> > > ### Author Response · Authors · 2023-11-22
> > > **Response**
> > >
> > > We thank the reviewer for the response and increasing the score!
> > >
> > > Re the medical image example: The initial state distribution would be a distribution of patients (with their actual underlying condition at the initial time step), the latent state at time $t$ represent the condition of their disease at that time, and an episode is the entire treatment process of this patient along the time horizon.

---

### Official Review · Reviewer_FWjL · 2023-11-04

**Soundness:** 4 excellent
**Presentation:** 3 good
**Contribution:** 3 good
**Rating:** 6
**Confidence:** 4

**Summary:**

Summary:

The papers consider learning in POMDPs, where in addition to receiving a single observation along the trajectory as is the case for standard POMDP setting, the learner can also get K additional observations from the corresponding latent states at every timestep (but at the end of the trajectory). The authors consider structural assumptions under which learning with K-observations is statistically tractable and provide algorithms to tackle them. The main results are as follows:
1. First, the authors show via a lower bound that even under the K observation model, we need additional assumptions to make learning tractable.
2. Then they consider additional structural called multi-observation revealing POMDPs and distinguishable POMDPs under which statistically efficient learning is possible. The former considers a rank-type assumption on the observation matrix whereas the latter considers a separability assumption on the columns. The two assumptions are equivalent to each other up to polynomial factor blow-up in k.
3. They provide algorithms for efficient learning under the above assumptions.

**Strengths:**

1. A new framework to consider POMDPs, and get statistically efficient algorithms.
2. Easy-to-understand analysis. The paper is only 20 pages long which is rare in the modern RL theory literature. This is primarily because the approach heavily builds on the OMLE algorithm from Liu et. al. 2022a.
3. A complete set of results.

**Weaknesses:**

1. The considered approaches are only statistically efficient. Can the authors provide a discussion on the possibility of getting computationally efficient or oracle-efficient algorithms?
2. I am not yet convinced by the motivation for considering k-observation settings, or natural problem settings where one can get k-observations in hindsight at the end of the trajectory. Can the authors provide examples of settings where (a) one can get k-observations, but (b) we do not have the ability to reset to the latent state (or generative model)? My worry is that the ability to reset to the latent state needs knowledge of the latent state which is practically equivalent to hindsight observability (in the game playing example provided in the paper one seems to need knowledge of the latent state to reset).


Another related work to compare to: "Agnostic Reinforcement Learning with Low-Rank MDPs and Rich Observations", Dann et. al. 2021. Their algorithm seems to work for POMDP settings with low-rank dynamics - while the dependence on d is exponential, their work does not seem to require any lower bound on \alpha and thus could be applicable when d is small/constant but \alpha could be arbitrarily small. I am pointing this out because the authors seem to portray \alpha-distinguishable POMDPs as the largest class of POMDPs that could be solvable statistically efficiently, however,  Dann et. al. 2021 gives another example which is solvable under orthogonal assumptions.

**Questions:**

Apart from the ones listed in the weaknesses above. I have a few more questions:

1. Can the authors provide examples / settings where there is a computational or statistical separation between K = 1 and K > 1. I am guessing that proposition 3 already captures this, but can you please provide more intuition on why we can expect such separation.
2. Are there settings where one can get k-observations but only at the end of the episode? In particular, one needs to wait to terminate to get more observations.
3.  From what I understand, it is a feature of the algorithm that only needs full trajectory information to construct MLEs. Hence, there is no separation between getting k-observations in real-time or at the end of the episode. Is there a fundamental separation between the two settings?
4. What is the dependence on H in the sample complexity bound?

---

> ### Author Response · Authors · 2023-11-18
> **Response to Reviewer FWjL**
>
> We thank the reviewer for the valuable feedback on our paper. We respond to the specific questions as follows.
>
> > The considered approaches are only statistically efficient. Can the authors provide a discussion on the possibility of getting computationally efficient or oracle-efficient algorithms?
>
> Our algorithm OST is actually oracle-efficient given the POMDP planning oracle POP. It only makes $T$ calls to POP, and the rest of the algorithm runs in ${\rm poly}(H, S, A, O, T, k)$ time, as all the main operations (computing bonus, closeness testing) runs in polynomial time. We have added a remark about this (in the paragraph before Theorem 11) in our revision. Further, if POP has an efficient approximate implementation (such as the quasi-polynomial approximate planning algorithm of Golowich et al. (2022) for observable POMDPs), then the whole algorithm can be made (quasi-)efficient.
>
> Golowich, N., Moitra, A., & Rohatgi, D. (2022). Planning in observable pomdps in quasipolynomial time. arXiv preprint arXiv:2201.04735.
>
> Our other algorithm $k$-OMLE is indeed only statistically efficient. How to make these OMLE type algorithms computationally more efficient is an important open problem, which we would like to leave as future work.
>
> > I am not yet convinced by the motivation for considering k-observation settings… Can the authors provide examples of settings where (a) one can get k-observations, but (b) we do not have the ability to reset to the latent state (or generative model)?
>
> To give an example, consider a medical imaging problem, where we usually take multiple images per time step to measure the latent state (true status of the patient) that cannot be directly observed, and the problem is sequential (need to repeat this along the treatment process). In this problem, we have multiple observations at each state, but we don’t have the ability to reset.
>
> > … My worry is that the ability to reset to the latent state needs knowledge of the latent state which is practically equivalent to hindsight observability (in the game playing example provided in the paper one seems to need knowledge of the latent state to reset).
>
> We agree that in the game playing example, the ability to reset is similar to having access to the latent state. However, a key difference here is that only the *environment* has access to the latent state, and the *learner* can still only learn about it from the (multiple) observations. In game playing, multiple observations could still be qualitatively different from the latent state — For example, the latent state could be the true status/properties of the characters, whereas the observations could be like results of a battle that could be played multiple times by saving/loading.
>
> We also remark that, in our setting, the value of $k$ matters and encodes a cost of making observations, which also makes it perhaps more realistic than hindsight observability. Even though by taking large $k$ one gets very close to hindsight observable, we do count it as $k$ samples when counting the sample complexity (e.g. in the previous medical imaging example, taking more measurements at each time step costs more time and resource).
>
> > Another related work to compare to: "Agnostic Reinforcement Learning with Low-Rank MDPs and Rich Observations", Dann et. al. 2021. Their algorithm seems to work for POMDP settings with low-rank dynamics - while the dependence on d is exponential, their work does not seem to require any lower bound on \alpha and thus could be applicable when d is small/constant but \alpha could be arbitrarily small. I am pointing this out because the authors seem to portray \alpha-distinguishable POMDPs as the largest class of POMDPs that could be solvable statistically efficiently, however, Dann et. al. 2021 gives another example which is solvable under orthogonal assumptions.
>
> We thank the reviewer for pointing out this related work. Using the reduction in that paper, they give an $\exp(d)$-samples algorithm for learning the best *reactive* policy in any tabular POMDP without further structural assumptions. However, we feel reactiveness makes the problem very different from—and arguably easier than—the standard problem of learning the (overall) best policy (which is in-general history dependent) considered in this work.
>
> As another minor remark, we did not mean to portray \alpha-distinguishable POMDPs as the largest class of statistically efficiently solvable in the standard setting; it is rather under our enhanced feedback model ($k$-observation feedback). But if any of our statements sound like portraying such a claim, we would be happy to revise.

---

> ### Author Response · Authors · 2023-11-18
> **Response to Reviewer FWjL (cont'd)**
>
> > Can the authors provide examples / settings where there is a computational or statistical separation between K = 1 and K > 1. I am guessing that proposition 3 already captures this, but can you please provide more intuition on why we can expect such separation.
>
> Thanks for the insightful question. As mentioned, (taking $K=2$ as an example), Proposition 3 is already a result along this line, where it gives examples of POMDPs that are revealing with 2 observations but not 1 observation.
>
> Building on that example, we may be able to further obtain a formal statistical separation, i.e. construct a family of POMDPs that can be solved in polynomial samples with 2 observations, but not so with 1 observation. As an initial idea, we can consider a combination lock with three latent states '+', '-', and '0', where the emission of '0' is a linear combination of those of '+' and '-', so that the problem is not revealing at $K=1$ but revealing at $K=2$, similar as the example in Proposition 3. Further designing the transitions and rewards may yield a statistical separation where with $K=1$ observations it requires $\exp(H)$ samples to solve, but with $K=2$ we can do with polynomially many samples since it is a revealing POMDP. We will think about this further.
>
> > Are there settings where one can get k-observations but only at the end of the episode? In particular, one needs to wait to terminate to get more observations.
>
> If we understood correctly, the reviewer means to ask why *we* reveal k-observations at the end of episodes rather than at each step? We chose this merely to avoid a potential confusion about *policies*---Restricting the trajectory to still have 1 observation restricts the policy itself to still be a standard POMDP policy, and prevents it from utilizing more observations to do better. We’d like to keep the policy class to be still the standard POMDP policy class, and only use the $k-1$ observations to help the learning within this class, hence we chose this setting. Of course, learning a policy to make use of all $k$ observations is also a sensible problem (e.g. in the medical imaging example mentioned earlier), which we did not pursue in this paper.
>
> > From what I understand, it is a feature of the algorithm that only needs full trajectory information to construct MLEs. Hence, there is no separation between getting k-observations in real-time or at the end of the episode. Is there a fundamental separation between the two settings?
>
> Right, once the policy class is fixed (to be the standard POMDP policy class), for the algorithms it does not make a difference when these observations are received.
>
> > What is the dependence on H in the sample complexity bound?
>
> The ${\rm poly}(H)$ is $H^5$ in Theorem 5, $H^6$ in Theorem 6, and $H^6$ in Theorem 11. We have updated the proofs correspondingly in our revision to show these.

---

### Author Response · Authors · 2023-11-18
**Revision Uploaded**

We thank all reviewers again for their valuable feedback on our paper. We have incorporated the reviewers’ suggestions and uploaded a revised version of our paper. For clarity, all changes (other than typo fixes) are marked in blue.

Best,
Authors

---

### Meta-Review · Area_Chair_NTUm · 2023-12-12

**Metareview:**

This paper tackles the sample-efficiency challenge in learning Partially Observable Markov Decision Processes (POMDPs). Introducing the "multiple observations in hindsight" feedback model, it allows the collection of additional observations after each POMDP interaction, enabling sample-efficient learning. The proposed model is demonstrated to be effective for two new POMDP subclasses—multi-observation revealing POMDPs and distinguishable POMDPs—expanding the scope beyond revealing POMDPs and offering practical solutions for real-world scenarios like game playing.

While the reviewers pointed out several limitations of the paper, the authors' rebuttal addressed many of the concerns. The presentation of the work still needs some effort. We encourage the reviewers to rewrite the paper to clarify some of the technical details, and also incorporate the other comments from the reviewers. We accept the submission.

**Justification For Why Not Higher Score:**

N/A

**Justification For Why Not Lower Score:**

N/A

---

### Decision · Program_Chairs · 2024-01-16

Accept (poster)